# Tristetraprolin binding site atlas in the macrophage transcriptome reveals a switch for inflammation resolution

Vitaly Sedlyarov[1,†], Jörg Fallmann[2,†], Florian Ebner[1], Jakob Huemer[1], Lucy Sneezum[1], Masa Ivin[1], Kristina Kreiner[1], Andrea Tanzer[2], Claus Vogl[3], Ivo Hofacker[2,4,5,*] & Pavel Kovarik[1,**]

## Abstract

Precise regulation of mRNA decay is fundamental for robust yet not exaggerated inflammatory responses to pathogens. However, a global model integrating regulation and functional consequences of inflammation-associated mRNA decay remains to be established. Using time-resolved high-resolution RNA binding analysis of the mRNA-destabilizing protein tristetraprolin (TTP), an inflammation-limiting factor, we qualitatively and quantitatively characterize TTP binding positions in the transcriptome of immunostimulated macrophages. We identify pervasive destabilizing and non-destabilizing TTP binding, including a robust intronic binding, showing that TTP binding is not sufficient for mRNA destabilization. A low degree of flanking RNA structuredness distinguishes occupied from silent binding motifs. By functionally relating TTP binding sites to mRNA stability and levels, we identify a TTP-controlled switch for the transition from inflammatory into the resolution phase of the macrophage immune response. Mapping of binding positions of the mRNA-stabilizing protein HuR reveals little target and functional overlap with TTP, implying a limited co-regulation of inflammatory mRNA decay by these proteins. Our study establishes a functionally annotated and navigable transcriptome-wide atlas (http://ttp-atlas.univie.ac.at) of cis-acting elements controlling mRNA decay in inflammation.

**Keywords** mRNA decay; inflammation; macrophage; PAR-CLIP
**Subject Categories** Genome-Scale & Integrative Biology; Methods & Resources; RNA Biology
**Mol Syst Biol. (2016) 12: 868**

## Introduction

Regulated mRNA decay has a fundamental impact on gene expression and is indispensable for life in metazoa (Stumpo *et al*, 2004, 2009; Ghosh *et al*, 2009; Katsanou *et al*, 2009; Hodson *et al*, 2010). mRNA decay is of central importance for timing and extent of inflammatory responses and for avoiding damaging hyperinflammation (Kafasla *et al*, 2014). Gene-targeted animals have revealed immune system-specific functions of a number of RNA-stabilizing and RNA-destabilizing proteins (Taylor *et al*, 1996; Lu *et al*, 2006; Matsushita *et al*, 2009; Pratama *et al*, 2013; Vogel *et al*, 2013). The molecular mechanisms that determine how these proteins select their targets and how they regulate temporal variation in target mRNA stability remain incompletely understood.

Tristetraprolin (TTP, gene name *Zfp36*) is one of the most significant immune system-specific mRNA-destabilizing proteins (Blackshear, 2002). TTP contains tandem CCCH-type zinc fingers, which mediate binding to 3′ untranslated regions (UTRs) of mRNA, predominantly at AU-rich elements (AREs) (Worthington *et al*, 2002; Blackshear *et al*, 2003). Binding of TTP to mRNA facilitates the recruitment of the CCR4-NOT deadenylase complex and the decapping complex (Fenger-Gron *et al*, 2005; Fabian *et al*, 2013). Recruitment of these complexes is followed by the degradation of the associated mRNAs within the microenvironment of processing bodies (Kedersha *et al*, 2005; Franks & Lykke-Andersen, 2007). TTP targets a number of inflammation-associated mRNAs for degradation, most notably those of cytokines and chemokines (Lai *et al*, 2006; Emmons *et al*, 2008; Stoecklin *et al*, 2008; Kratochvill *et al*, 2011; Rabani *et al*, 2014). In agreement with this selective mRNA targeting, TTP-knockout mice are born healthy but after several weeks of life develop severe and eventually lethal inflammation of multiple organs (Taylor *et al*, 1996). The inflammatory phenotype can be transferred by transplantation of TTP-deficient bone marrow

1  Max F. Perutz Laboratories, University of Vienna, Vienna, Austria
2  Institute for Theoretical Chemistry, University of Vienna, Vienna, Austria
3  Institute of Animal Breeding and Genetics, University of Veterinary Medicine Vienna, Vienna, Austria
4  Research Group Bioinformatics and Computational Biology, Faculty of Computer Science, University of Vienna, Vienna, Austria
5  Center for non-coding RNA in Technology and Health, University of Copenhagen, Frederiksberg C, Denmark
   *Corresponding author. Tel: +43 1427754608; E-mail: ivo.hofacker@univie.ac.at
   **Corresponding author. Tel: +43 1427752738; E-mail: pavel.kovarik@univie.ac.at
   †These authors contributed equally to this work

into wild type animals, but the specific hematopoietic lineage giving rise to the pleiotropic inflammation has not been identified (Carballo *et al*, 1997). Mice with LysMcre-driven TTP deletion, which ablates TTP in major myeloid cell types, are healthy despite an increased susceptibility to endotoxin shock (Kratochvill *et al*, 2011; Qiu *et al*, 2012).

TTP expression under steady state is ubiquitous and low, but it is strongly induced both transcriptionally and post-transcriptionally by inflammatory stimuli (e.g. bacterial products) or cytokines such as interferons, IL-4, and IL-10 (Mahtani *et al*, 2001; Suzuki *et al*, 2003; Sauer *et al*, 2006; Schaljo *et al*, 2009). TTP destabilizes its own mRNA, thereby providing a way for a rapid autoinhibitory feedback regulation after the disappearance of stimulatory cues (Tchen *et al*, 2004). Phosphorylation of TTP by the p38 MAPK and MK2 pathway stabilizes the otherwise proteasome-sensitive TTP, but it negatively regulates the mRNA-destabilizing activity of TTP (Stoecklin *et al*, 2004; Brook *et al*, 2006; Hitti *et al*, 2006; Clement *et al*, 2011).

The selective targeting of inflammation-associated mRNAs by TTP for degradation remains largely unexplained: (i) Although TTP preferentially binds AREs, most mRNAs containing AREs in their 3′ UTRs are stable; (ii) many ARE-containing mRNAs are destabilized independently of TTP; (iii) TTP can destabilize mRNAs lacking AREs (Raghavan *et al*, 2002; Yang *et al*, 2003; Lai *et al*, 2006; Stoecklin *et al*, 2008; Kratochvill *et al*, 2011). Analysis of TTP binding sites using PAR-CLIP in HEK293 cells overexpressing TTP confirmed preferential binding to AREs, but because of the largely missing expression of genes related to immune responses and the lack of immune signaling in HEK293 cells the study did not address TTP binding and its regulation in the most relevant context (Mukherjee *et al*, 2014). To understand TTP-dependent mRNA decay and its function, a precise knowledge of the positions and sequences of TTP binding sites in the transcriptome of immune cells in the context of a natural immune response is needed.

In the present study, we employed a modified PAR-CLIP method to identify at nucleotide resolution of TTP binding sites in the macrophage transcriptome in both inflammatory and resolution phases of responses to LPS. By combining this approach with transcriptome-wide gene expression and mRNA decay analyses, we establish a functionally annotated landscape of TTP-bound positions and sequences controlling mRNA stability during the inflammatory response. To account for combinatorial effects of mRNA-stabilizing and mRNA-destabilizing proteins, we similarly mapped binding sites of the mRNA-stabilizing protein HuR, which is often regarded as TTP antagonist (Srikantan & Gorospe, 2012; Tiedje *et al*, 2012). Our analysis is enhanced by RNA structure modeling of regions at and around TTP and HuR binding sites to identify the determinants of target selection. Together, our data and the accompanying navigation tools at the TTP atlas Web site (http://ttp-atlas.univie.ac.at) provide for the first time insights into the regulatory network of cis- and trans-acting factors controlling mRNA stability in the inflammatory transcriptome. The atlas revealed that TTP-directed mRNA decay is essential for entrance into the resolution phase rather than for the modulation of the onset of inflammation. Furthermore, our datasets demonstrated a largely nonoverlapping regulation of mRNA stability by TTP and HuR.

# Results

## PAR-iCLIP analysis in immunostimulated primary macrophages determines TTP binding landscape in the inflammatory transcriptome

To identify TTP binding positions and sequences, we employed PAR-iCLIP (**p**hoto**a**ctivatable **r**ibonucleoside-enhanced **i**ndividual nucleotide resolution **c**ross**l**inking and **i**mmunoprec**ip**itation) (Hafner *et al*, 2010; Konig *et al*, 2010; Zhang & Darnell, 2011) using specific TTP antibodies to target endogenous TTP in primary murine bone marrow-derived macrophages (BMDMs). BMDMs were stimulated for 6 h with LPS to establish an inflammatory gene expression pattern containing physiological TTP target RNAs in relevant quantities (Hao & Baltimore, 2009; Kratochvill *et al*, 2011). This approach allowed us to map TTP binding positions in the environment of stimulated immune cells, that is, a physiologically relevant system for interactions of TTP with RNA and for signal-regulated TTP activity (Stoecklin *et al*, 2008; Kratochvill *et al*, 2011). Notably, this strategy avoided both overexpressing TTP, which could result in the loss of binding specificity, and the use of cell types poorly expressing physiological TTP targets (e.g. chemokine and cytokine mRNAs).

To achieve crosslinking of TTP to juxtaposed RNA under mild conditions (365-nm UV light), RNA was metabolically labeled using thiouridine (4sU) which is readily utilized by BMDMs (Weintz *et al*, 2010). First, we tested the ability of the TTP antibody (Kratochvill *et al*, 2011) to specifically immunoprecipitate TTP-bound RNA in PAR-iCLIP experiments. Cell lysates from BMDMs derived from WT mice or mice lacking TTP in myeloid cells (ΔM mice) (Kratochvill *et al*, 2011) were subjected to a limited digestion using RNase I followed by immunoprecipitation. Subsequently, a $^{32}$P-labeled RNA linker was ligated to the immunoprecipitated RNA fragments to enable the detection of precipitated protein–RNA complexes after electrophoretic separation and blotting. RNA was detected in immunoprecipitations from WT, but not ΔM BMDMs, confirming the specificity of the TTP antibody (Fig EV1A). RNA fragments were then eluted from the membrane-bound RNA–protein complexes with proteinase K digestion and reverse-transcribed using primer complementary to the RNA linker.

UV-induced crosslinking of 4sU-labeled RNA causes the termination of reverse transcription or T to C (TC) transitions at the position of the protein–RNA crosslink (Hafner *et al*, 2010; Konig *et al*, 2010). The positions of reverse transcription termination or TC transitions serve as nucleotide resolution marks of protein–RNA crosslinking events in the subsequent analysis. Deep sequencing of transcriptome-wide PAR-iCLIP experiments revealed the accumulation of T at the position 0 (i.e. one nucleotide upstream of 5′ ends of sequencing reads), which is in agreement with reverse transcription termination only at positions of a crosslinked 4sU (Fig EV1B). The increased frequency of A and T as compared to G and C around the position 0 (Fig EV1B) reflects the high occurrence of A- and T-rich sequences (e.g. AU-rich elements) in TTP target sites (Fig 1). We detected a large number of TC transitions (44.9% of all substitutions) in the PAR-iCLIP reads (Fig EV1C), indicating that a protein–RNA crosslink caused, in addition to reverse transcription termination, frequent TC

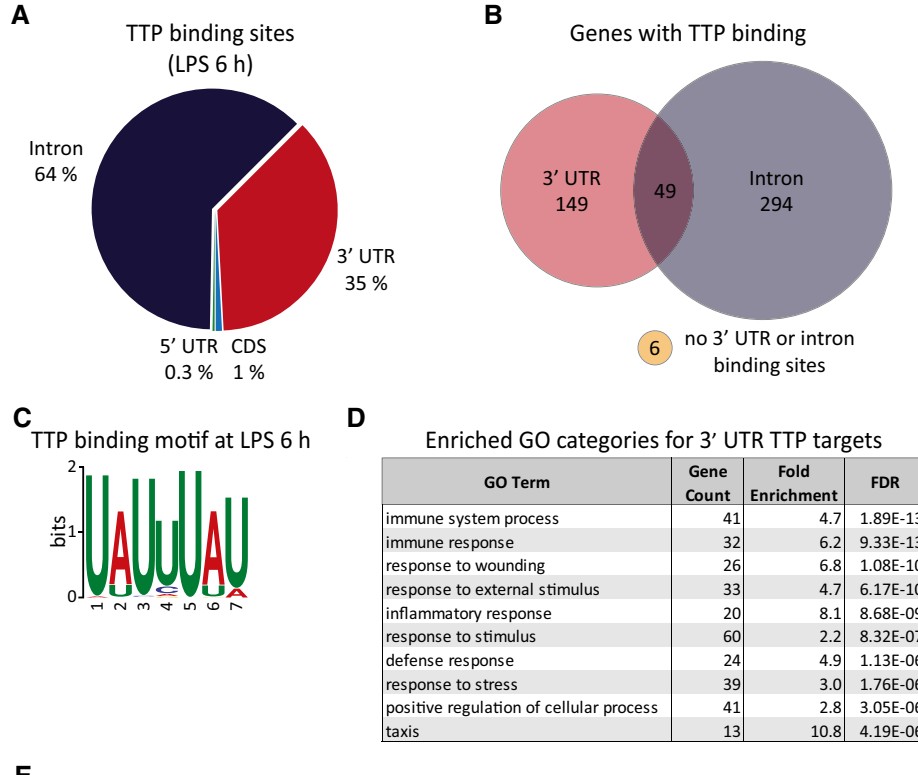

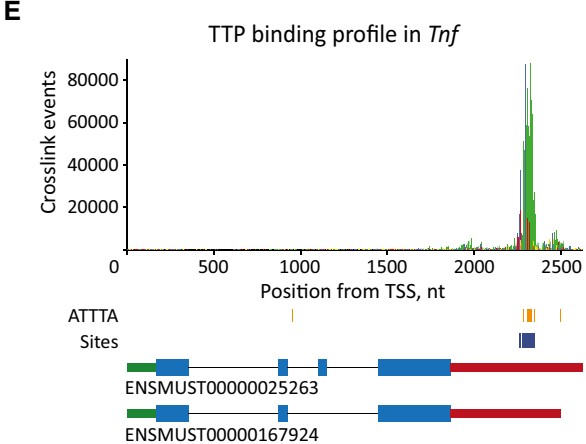

**Figure 1.  Introns and 3′ UTRs in the macrophage transcriptome are preferred TTP binding regions.**

A  Distribution of TTP binding sites in different transcript regions at LPS 6 h.
B  Number of genes with TTP binding mapped to 3′ UTR (red), intron (blue), both (overlap), or other regions (yellow).
C  Sequence logo representing the most abundant TTP binding motif at LPS 6 h ($E$-value $2.9 \cdot 10^{-1512}$).
D  Gene ontology enrichment analysis of genes with TTP binding in 3′ UTR (198 genes). GO biological process groups are sorted according to FDR-corrected $P$-value.
E  TTP binding in annotated *Tnf* transcripts. Transcript models are represented by green rectangles (5′ UTR), cyan rectangles (CDS), red rectangles (3′ UTR), and connecting lines (introns). Locations of AUUUA pentamers are underlined with orange rectangles. Detected TTP binding sites are underlined with dark blue rectangles (Dataset EV1). TSS, transcription start site; nt, nucleotide.

transitions during the reverse transcriptase read-through, as reported previously (Hafner *et al*, 2010). Furthermore, we observed a strong overlap between the positions of reverse transcriptase termination (i.e. positions 0 of reads) and TC transitions (Fig EV1D and E). Almost half (48%) of termination positions are at positions that harbor a TC transition in at least one other read (Fig EV1D). If one considers only termination positions within high-confidence TTP binding sites (see Materials and Methods), this percentage increases from 48 to 80% (Fig EV1E).

Control RNA-Seq experiments (i.e. samples without crosslinking) using 4sU-labeled rRNA-depleted RNA from LPS-treated BMDMs showed no bias for any nucleotide at the position 0 in RNA-Seq reads (Fig EV1F). Furthermore, the frequency of TC transitions in 4sU-labeled RNA was the same as that of other

substitutions. Thus, 4sU did not impair transcription fidelity in BMDMs or reverse transcription accuracy in samples from BMDMs not exposed to UV crosslinking (Fig EV1G). Together, these data confirmed that the conditions for 4sU labeling and UV crosslinking, as well as the use of the selected TTP antibody, resulted in specific crosslinking and immunoprecipitation of TTP-bound RNA in BMDMs.

Three independent PAR-iCLIP replicates were performed, and they produced a highly consistent number of mapped reads (Fig EV2A). To identify TTP binding sites in the datasets of crosslink positions (i.e. 5′ ends of reads), we used the Pyicos modFDR peak finding algorithm (Althammer et al, 2011), which determines TTP binding sites as regions with significantly higher number of crosslinks than would be expected by chance, thus distinguishing true signal from noise. The list of peaks derived from the Pyicos modFDR method was then filtered for peaks with a minimum of 100 crosslinks at their summit, generating only high-confidence TTP binding sites. Pyicos analysis carried out for all three replicates individually revealed a high correlation of the identified TTP binding sites among the replicates (Fig EV2A). This is exemplified by TTP binding to the Tnf transcript: TTP binding site was centered at the nucleotide (nt) 2,315 in the transcript (corresponding to nt 1,326 in mRNA) in all replicates (Figs 1 and EV2B), in agreement with the reported TTP binding to Tnf reporter constructs (Lai et al, 1999). TTP binding sites present in all three replicates (i.e. consensus binding sites) were used for further analysis. We identified 1,587 nonoverlapping TTP binding sites that were mapped to 498 protein-coding genes (Dataset EV1). The highest number of TTP binding sites were located in introns (64%) and 3′ UTRs (35%), whereas only 1.3% of binding sites were found in other exonic regions: 1% mapped to coding sequences (CDS) and 0.3% mapped to 5′ UTRs (Fig 1A). The high frequency of intronic binding was in agreement with the nearly equal distribution of normalized PAR-iCLIP signal between introns and 3′ UTR: The number of PAR-iCLIP reads divided by the number of RNA-seq reads in the corresponding feature, that is, introns or 3′ UTR, revealed a ratio of intronic to 3′ UTR PAR-iCLIP signal 1:0.98. Most target RNAs exhibited TTP binding either in introns (59%) or in 3′ UTRs (30%), while only a small fraction (10%) of the targeted RNAs displayed TTP binding in both introns and 3′ UTRs (Fig 1B). To determine the sequence motifs enriched within TTP binding sites, we employed MEME (Bailey & Elkan, 1994). The most abundant motif (E-value $2.9 \cdot 10^{-16}$) was a heptamer comprising the sequence UAUUUAU (Fig 1C), in agreement with previously described motifs (Mukherjee et al, 2014).

A predominant binding of TTP to introns and 3′ UTRs was recently found in a high-resolution mapping analysis of TTP binding sites in the human HEK293 cell line (Mukherjee et al, 2014). We thus asked whether the TTP binding sites identified by us in immunostimulated murine macrophages were related to those found in the HEK293 cells. Out of 4,625 total binding sites of the HEK293 dataset, we were able to assign 2,731 to homologous loci within annotated mouse genes. However, out of these 2,731 sites, only 32 positions in 27 genes overlap with TTP binding sites from our dataset (Table EV1). Notably, Tnf was not found among the targets in HEK293 cells, even though it is the most common TTP target in various cells. Other cytokine mRNAs were similarly absent (Table EV1). These dissimilarities might be caused by TTP overexpression, and hence a reduced binding specificity, in the HEK293 system. Furthermore, HEK293 cells are not immune cells and barely express genes related to the immune response. They also lack TTP activity regulated by immune cell signaling. Thus, the paucity of natural TTP targets in HEK293 cells and/or different TTP regulation might cause binding of TTP to RNAs different from those targeted in immune cells. Together, TTP binding in immunostimulated primary mouse macrophages considerably differs from binding in human HEK293 cells.

Binding of TTP to targets associated with immune responses of BMDMs was corroborated by gene ontology (GO) analysis: mRNAs targeted by TTP at 3′ UTR (198 genes) were significantly enriched in genes involved in various immune processes (Fig 1D). Previously reported TTP targets in mouse macrophages including Tnf, Il1a, Il1b, Il6, Il10, Cxcl1, Cxcl2, Ccl3, or Ccl4 were all found in our collection of mRNAs containing TTP binding sites (Dataset EV1). Importantly, the TTP binding sites correspond well to positions characterized by in vitro binding or reporter assays in earlier studies: Cxcl1 (Datta et al, 2008), Il1a and Cxcl2 (Kratochvill et al, 2011), and Ccl3 (Kang et al, 2011) (Table 1).

In summary, by employing PAR-iCLIP analysis in immunostimulated macrophages, we established a comprehensive collection of high-confidence positions and sequences bound by TTP in the inflammatory transcriptome of the macrophage.

## TTP binding to 3′ UTR is an essential but not sufficient requirement for target mRNA destabilization in immunostimulated macrophages

TTP destabilizes a number of inflammation-associated mRNAs often containing AREs in their 3′ UTRs, but the mechanism of selective mRNA destabilization by TTP has remained elusive (Lai et al, 2006; Emmons et al, 2008; Stoecklin et al, 2008; Kratochvill et al, 2011). In particular, it has been unclear whether stable mRNAs containing potential TTP binding sites are bound by TTP, but remain stable in the context of immune cell signaling, or whether such mRNAs are not accessible to binding by TTP under such conditions.

**Table 1. Positions of TTP binding sites from previous publications and PAR-iCLIP experiment.**

| TTP target | Positions of reported binding sites (nt) | References | Positions of binding sites determined by PAR-iCLIP (nt) |
|---|---|---|---|
| Ccl3 | 515–524 | Kang et al (2011) | 500–594 |
| Cxcl1 | 449–468 | Datta et al (2008) | 414–440; 445–457; 462–471 |
| Cxcl2 | 499–527 | Kratochvill et al (2011) | 481–487; 490–536; 1,035–1,037 |
| Il1a | 1,458–1,492 | Kratochvill et al (2011) | 1,423–1,491; 1,501–1,503 |

                                

To explore the effects of TTP binding on target transcript stability in activated macrophages, we determined mRNA decay rates in the transcriptome of WT and ΔM macrophages stimulated for 6 h with LPS (Dataset EV2), that is, under the same conditions as used for our PAR-iCLIP analysis, and examined the stability of transcripts containing TTP binding sites. We analyzed genes containing TTP binding sites in 3′ UTRs separately from those containing binding sites only in introns. Genes exhibiting TTP binding in both 3′ UTR and introns were included in both categories. In the 3′ UTR category, we observed that 14% of mRNAs (28 genes) were unstable in a TTP-dependent way, 15% (29 genes) were unstable independently of TTP, but a remarkably large number of mRNAs (71%, 141 genes) were stable (Fig 2A and Dataset EV3). Importantly, the length and the position of binding sites as well as their normalized scores (crosslink events normalized to expression, see Materials and Methods) were comparable for both stable mRNAs (e.g. *Sdc4*) and mRNAs destabilized by TTP (e.g. *Il6)* (Fig 2C and D). In the category of transcripts bound by TTP in introns, the vast majority (74%; 252 genes) of the corresponding mRNAs were stable, and only 1%

(three genes) were destabilized in a TTP-dependent manner (Fig 2B and Dataset EV4). Mukherjee and colleagues found no effects of intronic TTP binding on the steady-state levels of the corresponding mRNAs HEK293 cells (Mukherjee *et al*, 2014). Our study advances this report in that it directly shows that binding of TTP to introns does not in general cause destabilization of the corresponding mRNAs.

Collectively, the results demonstrate that the mRNA-destabilizing function of TTP in macrophages is almost completely restricted to transcripts bound by TTP in 3′ UTRs. However, binding of TTP is not sufficient for destabilization of the targeted transcript since many 3′ UTR-bound mRNAs in the macrophage transcriptome are stable.

### TTP binding to introns does not interfere with processing of target RNA

To further characterize intronic binding of TTP, we examined in detail binding of TTP to the intron 4 of *Irg1* (Fig 3A), the highest

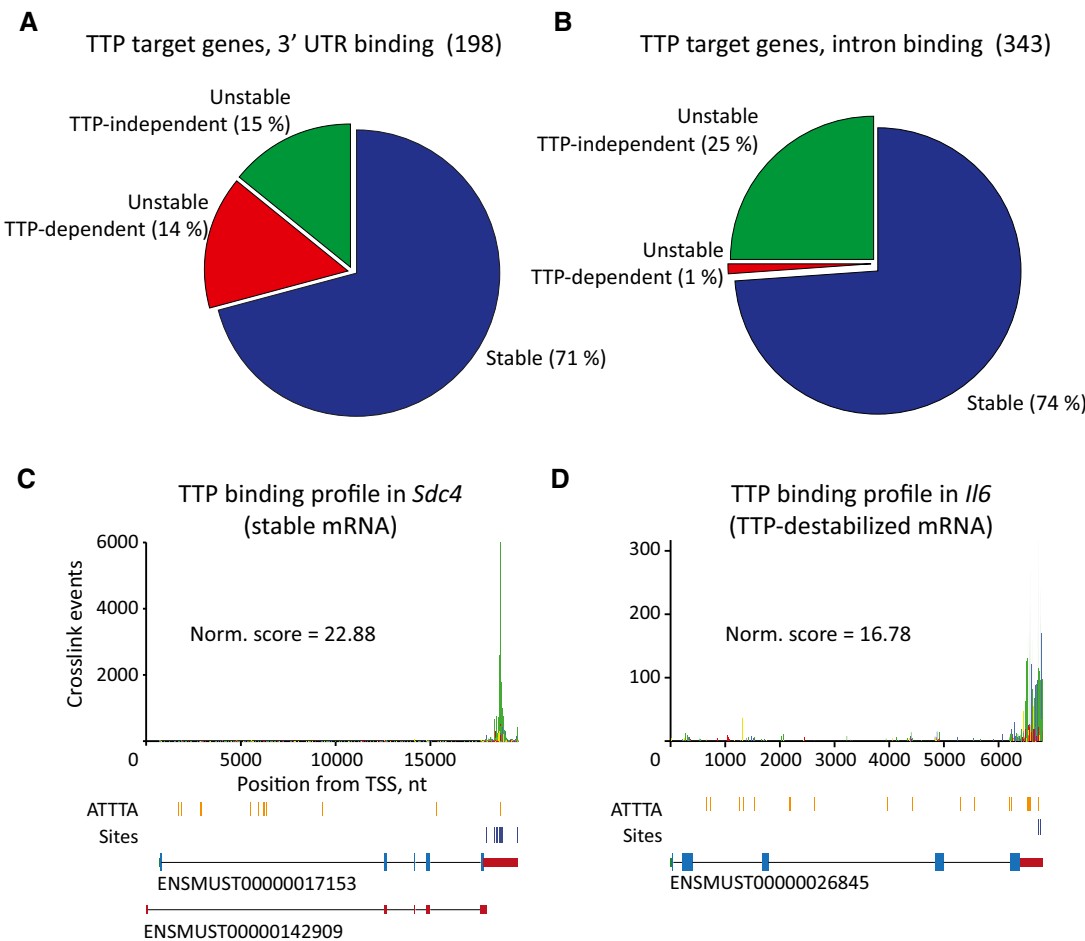

**Figure 2. TTP binding to 3′ UTR is essential but not sufficient requirement for target destabilization.**

A  Stability of mRNAs targeted by TTP in 3′ UTR.
B  Stability of mRNAs containing TTP binding sites in introns of the corresponding pre-mRNAs. Transcripts targeted by TTP in both 3′ UTRs and introns are included in both (A) and (B).
C  TTP binding profile in the *Sdc4* transcript as an example of stable mRNA bound by TTP.
D  TTP binding profile in the *Il6* transcript as an example of mRNA destabilized by TTP.

  

**Figure 3.   TTP binds to spliced-out intron of *Irg1* without influencing *Irg1* transcript processing.**

A   TTP binding profile in the *Irg1* transcript (upper panel) and detailed view of the binding region (lower panel). Each vertical bar represents the number of crosslink events in the corresponding position. The bars are color-coded according to the nucleotide at the corresponding position.

B   *Irg1* mRNA expression profile in BMDMs stimulated with LPS, normalized to *Hprt* (AU, arbitrary units). Error bars represent 95% confidence interval, $n$ = 3 biological replicates.

C   *Irg1* mRNA stability assay. BMDMs were stimulated for 3 h (left panel) or 9 h (right panel) with LPS and transcription was stopped by actinomycin D (ActD) followed by measurements of remaining mRNA 30 and 60 min after the transcription blockage. *Irg1* mRNA is stable (half-life > 180 min) after 3 and 9 h of LPS stimulation in both WT and TTP-deficient BMDMs.

D   Expression profile of intron 3 (I3) (left panel) and the TTP binding site containing intron 4 (I4, right panel) of *Irg1*, normalized to *Hprt*. Error bars represent 95% confidence interval, $n$ = 3 biological replicates.

E   *Irg1* read coverage in RNA-Seq experiments for WT and ΔM BMDMs. *Irg1* gene (ENSMUSG00000022126) has a single annotated transcript (ENSMUST00000022722).

F   TTP binds to spliced-out intron 4. Data depict RNA-IP experiments showing that TTP binds to intron 4 (I4) but not to the pre-mRNA containing exon 4 and intron 4 (E4-I4) (upper panel). PCR design for the detection of cDNA corresponding to intron 3, intron 4, exon 4/intron 4 is shown in lower panel. Binding to *Tnf* mRNA was used as a positive control. Data are presented in arbitrary units (AU) as $2^{25-Ct}$.

G   Western blot showing stable amounts of nuclear TTP during LPS stimulation for indicated times. Tubulin and histone H3 were used as controls for the successful separation of cytoplasmic (tubulin) from nuclear (histone H3) fractions, respectively.

Source data are available online for this figure.

scoring intronic target and overall the fourth best scoring target (Datasets EV1 and EV4). *Irg1* mRNA levels were low but strongly induced by LPS to approximately 1.5-fold higher levels in WT compared to ΔM BMDMs (Fig 3B). The *Irg1* mRNA was stable in both WT and ΔM BMDMs 6 h after LPS stimulation (Dataset EV4) as well as at other time points (Fig 3C). To test whether the higher *Irg1* mRNA levels in WT cells resulted from increased transcription, we analyzed primary transcript levels in the nuclear RNA fraction by qPCR for intron 3 (an intron not targeted by TTP) and intron 4 (the intron comprising the TTP site). Similar to total mRNA levels, *Irg1* primary transcript levels at both introns were higher in WT than in ΔM BMDMs (Fig 3D). Intronic TTP binding did not influence the abundance of intronic RNA because the ratios of both TTP-targeted and TTP-nontargeted *Irg1* introns were similar in WT and ΔM BMDMs and comparable to the ratio of the mature mRNAs. Importantly, the *Irg1* read coverage in WT cells was similar across the entire gene to the profile in ΔM BMDMs (Fig 3E), confirming that TTP did not regulate *Irg1* transcript processing. To examine whether TTP associates with the *Irg1* intron prior to or after splicing, 4sU-labeled and UV-crosslinked RNA was immunoprecipitated from LPS-treated WT and ΔM BMDMs using TTP antibodies. In contrast to PAR-iCLIP, the RNA was not digested with RNase I so that the full-length TTP-bound RNA molecules were immunoprecipitated. Quantification of intron 3 and intron 4 confirmed that TTP binds only to the latter (Fig 3F). We could not detect RNA spanning the junction between exon 4 and intron 4 (Fig 3F), suggesting that TTP was bound to the spliced-out intron.

To more broadly assess whether TTP binding to introns can regulate pre-mRNA processing, we compared relative expression (i.e. fragments per kilobase of exon per million fragments mapped, FPKMs) of the transcript isoforms of 10 highest scoring intronic TTP targets in WT and ΔM BMDMs (Dataset EV5). The abundance of transcript isoforms was in general similar in WT and ΔM BMDMs, indicating that TTP does not influence transcript processing.

The frequent binding of TTP to introns suggested that a certain amount of TTP is present in the nucleus of immunostimulated BMDMs. To confirm this, we fractionated BMDMs over the course of the inflammatory response and examined TTP amounts in the cytoplasmic and nuclear compartments. The experiment confirmed that the highest amount of TTP was in the cytoplasmic fraction, but

substantial levels were also observed in the nuclear fraction (Fig 3G).

In conclusion, binding of TTP to introns does not affect processing and splicing of the targeted transcript. The data also indicate that TTP can bind to spliced-out intronic RNA.

## TTP and HuR bind mostly to different mRNAs

The lack of detectable destabilization of many mRNAs bound by TTP prompted us to examine whether such mRNAs might also be targeted by HuR, a key mRNA-stabilizing protein with important functions in immune cells (Yiakouvaki *et al*, 2012). Positive effects of HuR on target RNA stability and/or translation can be caused by competition with TTP for binding to target RNA, as shown for *Tnf* mRNA in macrophages (Tiedje *et al*, 2012). Targeting of mRNAs by both proteins has been shown to occur also in HEK293 cells (Lebedeva *et al*, 2011; Mukherjee *et al*, 2011). However, a genome-wide HuR and TTP target overlap at physiological levels of both the target mRNAs and HuR and TTP remained unknown. To investigate such overlap and the potential for a combinatorial regulation of inflammatory mRNAs, we performed HuR PAR-iCLIP in BMDMs stimulated for 6 h with LPS, as described above for TTP. We precipitated endogenous HuR crosslinked to RNA using a specific HuR antibody (Fig EV3A). Two independent biological replicates produced consistent numbers of uniquely mapped reads which after Pyicos analysis revealed 2,380 HuR binding sites present on both replicates. These consensus binding sites mapped to 303 genes, mostly in 3′ UTRs (78%) followed by sites in introns (17%), 5′ UTRs (3.5%), and CDS (1.5%) (Fig 4A and Dataset EV6). GO analysis showed that 3′ UTR HuR targets were significantly enriched in genes involved in diverse processes ranging from immune responses to development, organelle organization, or metabolism (Table EV2). This rather pleiotropic functional assignment contrasts the predominantly immune response-associated processes found in GO analysis of TTP binding sites (Fig 1D), and MEME analysis of HuR binding sites revealed a U-rich nonamer UUUUUUUUU as most-overrepresented binding motif (*E*-value < $2.2 \cdot 10^{-16}$) (Fig 4B), similar to the motif previously identified in HEK293 cells (Lebedeva *et al*, 2011; Mukherjee *et al*, 2011).

We found two times more HuR than TTP binding sites per 3′ UTR, while their length was similar for both proteins (Fig 4C and

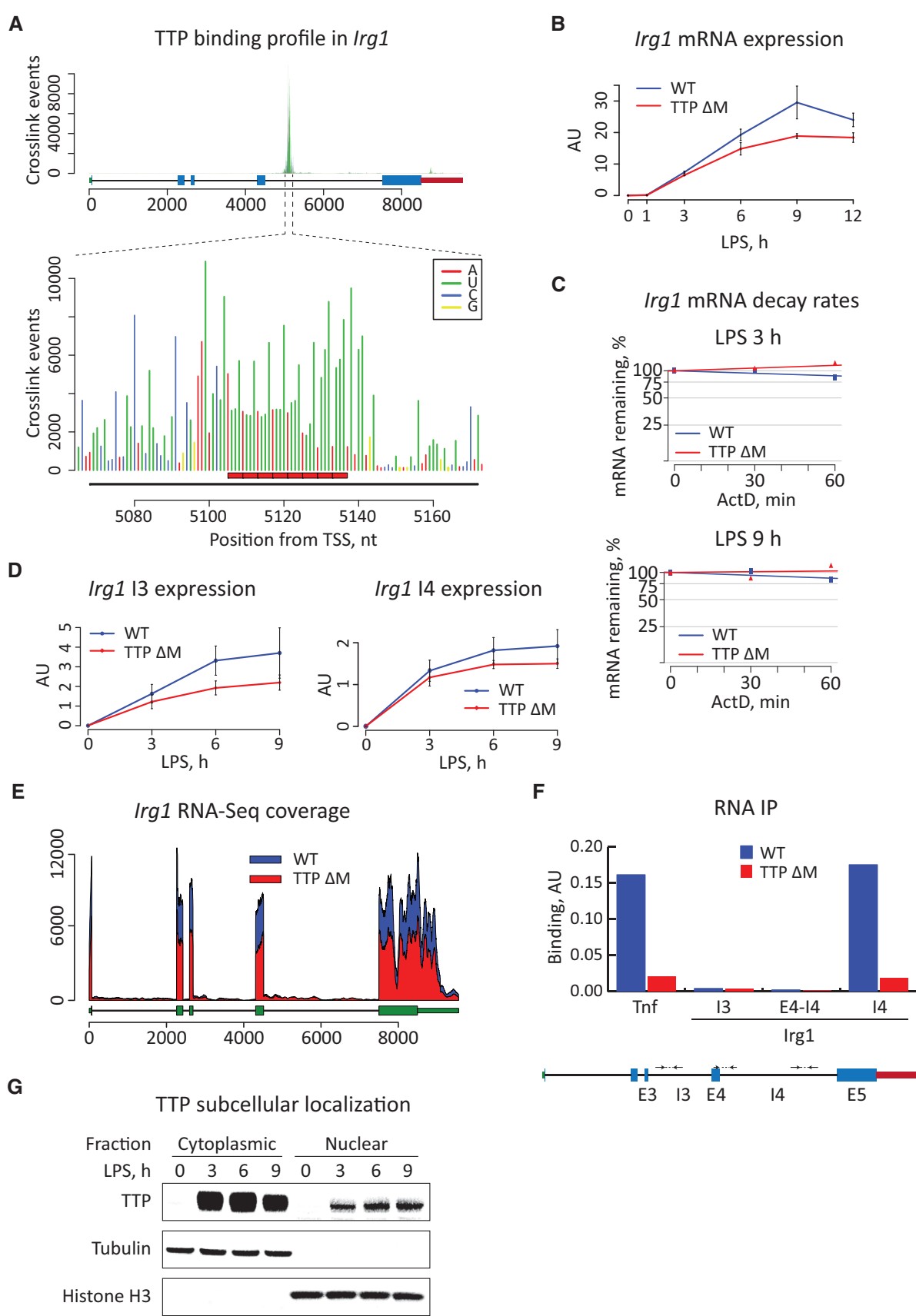

**Figure 3.**

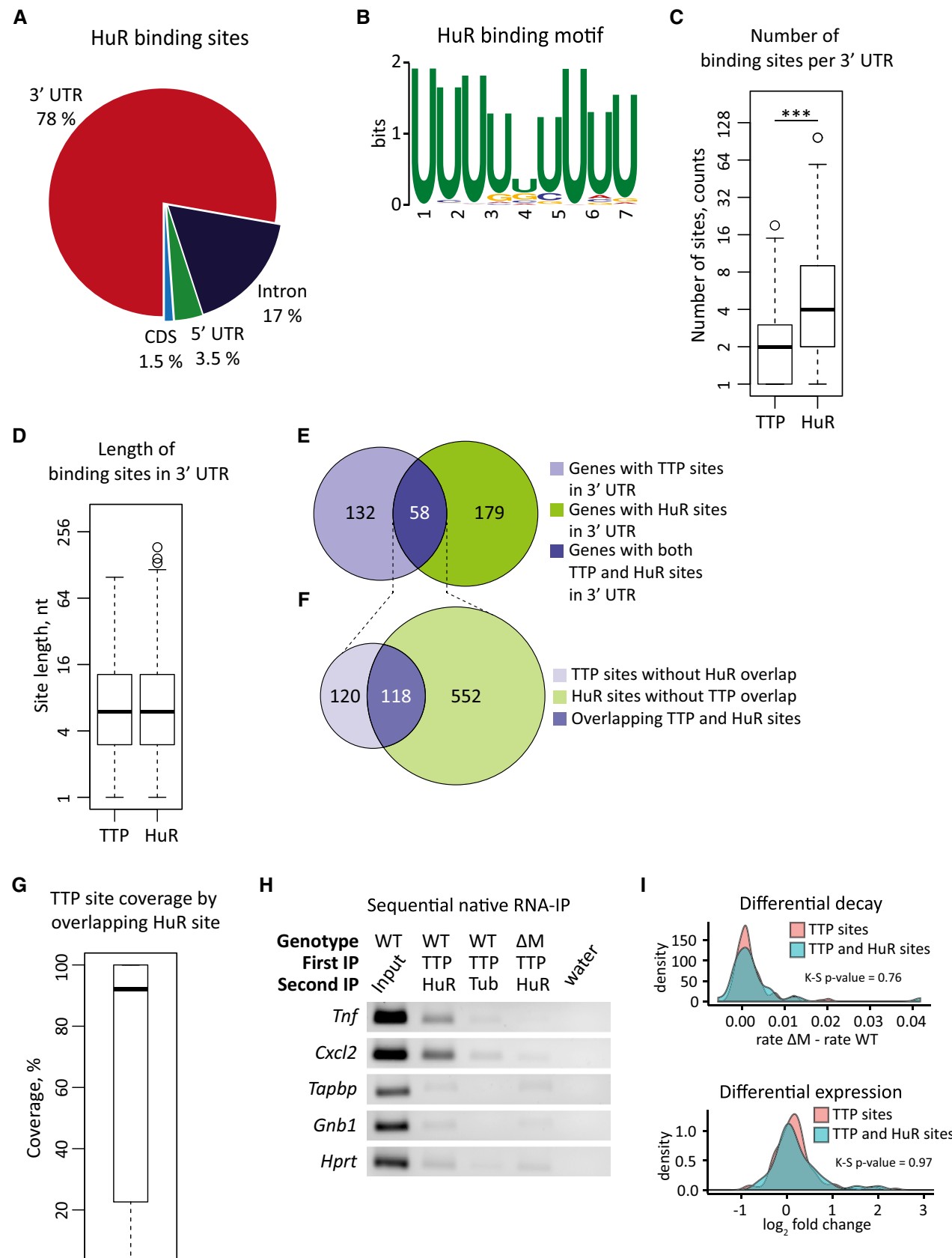

**Figure 4.**

◀

**Figure 4.  TTP and HuR target mostly different transcripts in the macrophage transcriptome.**

A   Distribution of HuR binding sites in different transcript regions.

B   Sequence logos representing the most abundant HuR binding motif at LPS 6 h.

C   Distribution (box-and-whisker plots) of the number of TTP and HuR binding sites per 3′ UTR. The number of sites is presented in logarithmic scale ($\log_2$). Median number of sites per 3′ UTR is 2 and 4 for TTP and HuR, respectively. The difference between samples is statistically significant ($P$-value = $1.2 \times 10^{-9}$, $\alpha$ = 0.05) as tested with Mann–Whitney *U*-test.

D   Distribution (box-and-whisker plots) of lengths of TTP and HuR binding sites. Median site length is 6 nt for both TTP and HuR sites.

E   The numbers of genes targeted at 3′ UTR by TTP (132) or HuR (179) or both (58). Genes with heterogeneous annotation (genes containing genomic features annotated as 3′ UTR and intron in different transcripts) were excluded from the analysis.

F   Mutual relationship of TTP and HuR binding positions within a group of mRNAs targeted by both proteins: 120 TTP and 552 HuR binding sites show no overlap and 118 TTP and HuR sites overlap by at least 1 nt.

G   Overlapping TTP and HuR binding sites in 3′ UTR exhibit large extent of overlap. Box-and-whisker plot for the distribution of coverage (in % of coverage) of a given TTP site by HuR sites in the group of targets containing overlapping binding sites (118 sites) reveals a median coverage of 92% (upper quartile—100%, lower quartile —23%). Note that 100% coverage corresponds to coverage of all nucleotides of a given TTP site by HuR site.

H   Sequential native RNA-IP from WT and ΔM BMDMs. Simultaneous binding of TTP and HuR to the same RNA molecule was tested with IP using anti-TTP antibodies (first IP) followed by elution with specific peptide and consecutive IP with anti-HuR antibodies (second IP). Second IP performed with anti-tubulin antibodies serves as a control for the specificity of second IP reaction. Sample with ΔM BMDMs was used as a control for the specificity of the TTP antibody.

I   Kernel density estimate plots for differential decay (upper panel) and differential expression (lower panel) in the group of genes with overlapping TTP and HuR sites (red) or TTP sites only (green). The difference in differential decay and differential expression between these two groups is not significant, as tested using Kolmogorov–Smirnov test (K-S).

Source data are available online for this figure.

D). The majority (84%) of target 3′ UTRs contained either HuR (179 genes) or TTP (132 genes) binding sites, while only 59 target genes exhibited binding sites for both HuR and TTP (Fig 4E). Within this set of 59 target genes, we identified 552 and 120 nonoverlapping HuR and TTP sites, respectively, and 118 overlapping sites (i.e. overlap by at least 1 nt) (Fig 4F). Most of the overlapping sites displayed an overlap over the entire TTP binding site (Fig 4G). The overlapping category comprised 40 genes, including *Tnf* (Fig 4H). The size of the mapped binding sites (Figs EV2B and EV3B) would allow both proteins to bind simultaneously. To test this possibility, we performed sequential RNA-IPs by TTP immunoprecipitation followed by peptide-mediated elution of TTP complexes and the subsequent HuR immunoprecipitation. Sequential RNA-IPs revealed that TTP and HuR can bind simultaneously to *Tnf* and *Cxcl2* mRNAs, that is, targets that contain overlapping binding sites (Fig 4H). No HuR binding was detected to *Tapbp* and *Gnb1* mRNAs, which contain HuR but not TTP binding sites (Datasets EV1 and EV3 and Fig 4H). This result corroborated the PAR-CLIP data and strengthened the hypothesis that mRNAs targeted by both TTP and HuR might be co-regulated by these two proteins. However, differential decay rates and differential

expression of mRNAs containing overlapping binding sites (Table 2) were not significantly different from mRNAs containing only TTP binding sites (Fig 4I). This indicated that mRNAs targeted by both TTP and HuR are, in general, not co-regulated by these two proteins at the level of mRNA stability. Such mRNAs might be rather co-regulated at the level of translation, as proposed for *Tnf* (Tiedje *et al*, 2012).

In summary, in contrast to expectations, our data reveal that co-regulation of mRNA stability by TTP and HuR in immunostimulated macrophages is rather an exception than a common principle.

### Structural context of target sites is more important for HuR than TTP binding

The limited number of mRNAs targeted by both TTP and HuR prompted us to examine the binding sites for both proteins in a structural and sequence context. To that purpose, we analyzed the structuredness of the region surrounding either bound or unbound ARE motifs. As a measure of structuredness, we computed accessibilities, that is, the probability that a given region remains unpaired, for all TTP/HuR target mRNAs using RNAplfold (Bernhart *et al*, 2011). We used a sliding window of W = 75 and then extracted accessibilities for 7-nt stretches, corresponding to the length of the WATTTAW TTP-ARE core motif, along bound and unbound regions, respectively. Both TTP and HuR motifs within binding sites show a higher probability of being unpaired than motifs without binding (Fig 5A and B). Moreover, bound motifs were typically embedded in a large (> 15-nt) unstructured and AU-rich region. To quantify how structuredness and AU content contributed to binding, we trained linear discriminators based on these features and performed receiver-operating characteristic (ROC) analysis. For TTP, AU content and structuredness were equally efficient at predicting bound motifs, while for HuR structuredness was a better predictor (Fig 5C and D). Thus, both TTP and HuR require the binding motifs to be embedded in accessible (unstructured) regions with high AU content for successful interactions, but the structure is more influential in the case of HuR.

**Table 2.   List of genes with overlapping TTP and HuR sites in 3′ UTR (40 genes).**

| | | | |
|---|---|---|---|
| 4933426M11Rik | Maf | Ccl4 | Prdx1 |
| Cdkn1a | Spp1 | Cxcl2 | Ywhaz |
| Lass6 | B2m | Pfkfb3 | Cd44 |
| Sdc4 | Cmpk2 | Trim30a | Il1rn |
| Actb | Marcks | Ccl9 | Ptgs2 |
| Cebpb | Tnf | Fth1 | Zeb2 |
| Lipg | Ccl3 | Pim1 | Cd47 |
| Sod2 | Cxcl10 | Txnrd1 | Irg1 |
| Arl8a | Nfkbia | Cd274 | Rsad2 |
| Cflar | Tor1aip1 | Hmox1 | Zfp36 |

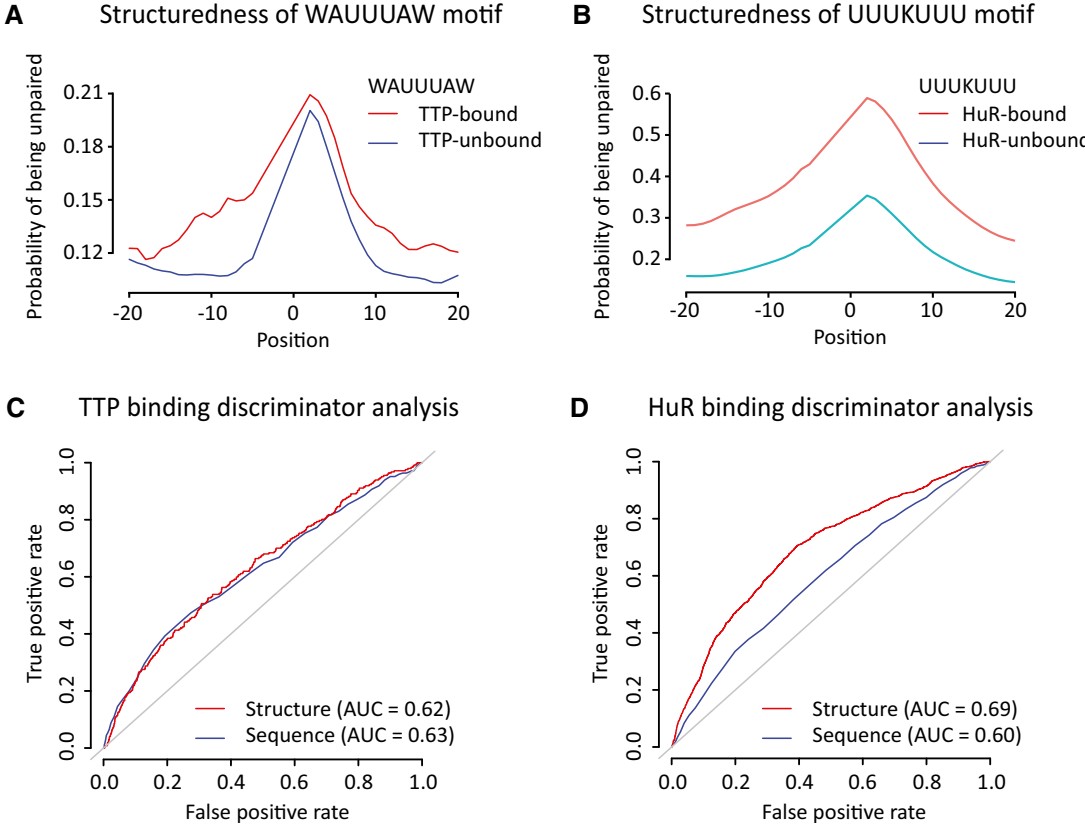

**Figure 5.  Structural context of target sites is more important for HuR than TTP binding.**

A   Regions containing the preferred TTP binding motif WAUUUAW show increasing probability of being unbound toward the central motif. Bound motifs are significantly less structured than unbound ones.

B   Regions containing the preferred HuR binding motif UUUKUUU have a high probability of being unpaired, with its peak at the center. This probability almost doubles for bound motifs.

C   Receiver-operating characteristic (ROC) analysis evaluating if sequence (AU content) or structure is a better classifier in linear discriminant analysis between bound and unbound TTP target motifs. Little difference between sequence (red, area under curve (AUC): 0.62) and structure (blue, AUC: 0.63) classifiers.

D   Same as (C), but for HuR bound and unbound target motifs. Structure classifier shows higher AUC (0.69) than AUC of sequence classifier (0.60).

**TTP target spectrum but not the binding motif shifts during the transition from the inflammatory to the resolution phase of the macrophage response**

TTP-mediated mRNA destabilization is known to be qualitatively and quantitatively regulated during the inflammatory response: Some mRNAs (e.g. *Tnf* mRNA) are destabilized by TTP throughout the inflammatory response, while other mRNAs (e.g. *Cxcl2* mRNA) are initially stable and become unstable in the resolution phase of inflammation (Tudor *et al*, 2009; Kratochvill *et al*, 2011). It remained unclear whether such a shift in target mRNA selection was caused by changes in TTP binding. To address this question, we mapped TTP binding sites at 3 h of macrophage stimulation with LPS by using the same approach as for the 6-h analysis described above. The 3- and 6-h time points represent the peak inflammatory phase and the beginning of the resolution phase, respectively: The expression of many important inflammatory mRNAs (e.g. *Tnf*, *Cxcl2*) culminates around 3 h and declines after 6 h of LPS treatment (Hao & Baltimore, 2009; Kratochvill *et al*, 2011).

Analysis of three PAR-iCLIP replicates at 3 h of LPS stimulation revealed consensus TTP binding sites that mapped to 465 genes (Dataset EV7). These sites were located mostly in introns and 3′ UTRs (Fig 6A), similar to the 6-h time point. We then focused on the analysis of 3′ UTR sites, since mRNA-destabilizing activity of TTP is confined to this category. The number of genes targeted by TTP increased between 3 and 6 h of LPS stimulation from 157 to 198 (by 27%) (Fig 6B). Over 80% of the 3-h targets were also found among the 6-h targets (Fig 6B). The increased TTP binding at 6 h of LPS stimulation was validated using native RNA-IP: Consistent with Datasets EV1 and EV7, TTP binding to *Tnf*, *Ccl3*, and *Zfp36* mRNAs did not change, whereas binding to *Fos* and *Ccl12* mRNAs strongly increased between 3 and 6 h of LPS treatment (Fig EV4). These data showed that a shift in TTP target selection occurred between the peak inflammatory phase and early resolution phase. Gene ontology (GO) analysis using GO categories significantly enriched at 3 and 6 h of LPS (false discovery rate (FDR) < 0.05, Table EV3) corroborated this finding: Although most targets belong to biological processes enriched at both time points (18 GO terms), many targets are in biological processes unique for

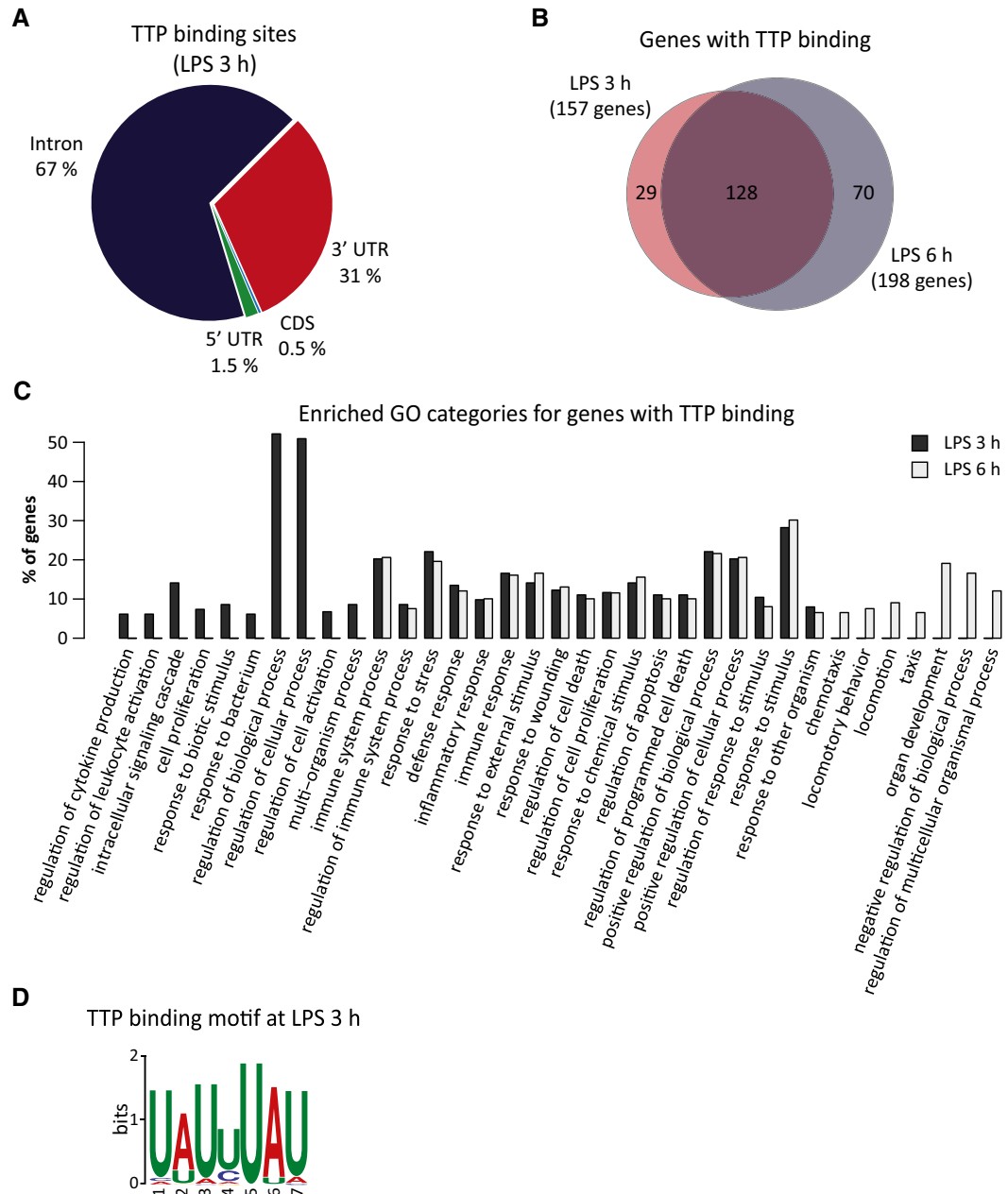

**Figure 6.  TTP target spectrum shifts during progression from the inflammatory to the resolution phase of macrophage response.**

A   Distribution of TTP binding sites in different transcript regions after 3 h of LPS treatment.
B   Overlap between TTP-bound mRNAs at 3 and 6 h of LPS treatment.
C   Enriched GO biological process categories for TTP-bound genes at 3 and 6 h of LPS treatment (enrichment FDR < 0.05).
D   Sequence logo representing the most abundant TTP binding motif at 3 h of LPS treatment.

each time point (10 and 7 unique GO terms at 3 and 6 h, respectively) (Fig 6C, Table EV3). Enriched GO terms common for both time points include the immune system process and immune response as top processes (Fig 6C). GO terms unique for the 3-h time point represent mostly activation of the immune response (e.g. regulation of cell activation), whereas GO terms unique for the 6-h time point include taxis, chemotaxis, and locomotion (Fig 6C), which represent the processes involved in the

maintenance of inflammation (Griffith *et al*, 2014). This shift in target selection was not caused by changing preferences in target site sequences since most over-represented motifs at both time points were very similar, that is, heptamers comprising the UAUUUAU sequence (Fig 6D).

Thus, TTP prefers similar target sequences in both inflammatory and early resolution phases, although the number of targeted mRNAs increases by 1/5 and the spectrum shifts to targets involved

in the perpetuation of inflammation as the cells progress through the response.

### TTP-dependent mRNA decay controls a switch for entering the resolution phase of the macrophage response

To functionally annotate the mapped TTP binding sites in immunostimulated macrophages, we wished to obtain transcriptome-wide gene expression data and mRNA decay rates in WT and ΔM macrophages under the same conditions as applied during our PAR-iCLIP experiments. We first determined differential gene expression using WT and ΔM macrophages stimulated for 0, 3, and 6 h with LPS to cover the onset of inflammation and the switch to the resolution phase. The expression of 488 genes was elevated (FDR < 0.05) in unstimulated macrophages lacking TTP (ΔM macrophages) as compared to WT cells (Dataset EV8 and Fig 7A). The number of stronger expressed genes in ΔM cells increased to 991 and 1,103 after 3 and 6 of LPS treatment (Dataset EV8) (Fig 7A).

Analysis of GO categories enriched in differentially expressed genes at 3 and 6 h of LPS (FDR < 0.05, Table EV4) confirmed a major involvement of TTP in regulating the immune response since the GO terms "immune response" and "immune system process" were the top-ranked processes (Fig 7B, Table EV4). However, similar to the GO analysis of TTP binding sites (Fig 6C), the GO terms "taxis" and "chemotaxis" characteristic for the perpetuation of inflammation were found only at the 6-h time point (Fig 7B).

We asked whether the progressive changes in differential expression are caused by similar changes in mRNA decay. To this end, we first determined mRNA decay rates in WT and ΔM macrophages at 3 h of LPS treatment (Dataset EV2), as described for the 6-h time point. Correlation analysis of mRNA decay rates at 3 and 6 h of LPS stimulation revealed that the impact of TTP-dependent destabilization increases with the time of LPS treatment (Fig 7C). This finding was validated for several targets using qRT–PCR: *Cxcl1, Ccl4, Il10,* and *Zfp36l2* mRNAs were more strongly destabilized at 6 h than at 3 h of LPS treatment, whereas *Irf1* mRNA was decaying similarly at both time points (Fig EV5), consistent with the Dataset EV2. Subsequently, Pearson's correlation coefficients for differential decay versus differential expression at 3 and 6 h of LPS stimulation were calculated. The analysis showed an increased influence of TTP-dependent mRNA decay on the expression profile at the transition to the resolution phase of the inflammatory response: Different decay rates between WT and ΔM cells influenced negligibly gene expression at 3 h (Pearson's correlation $\rho = 0.08$), but strongly at 6 h (Pearson's correlation $\rho = 0.33$) of LPS treatment (Fig 7D).

These data establish that TTP-mediated mRNA decay directly controls the switch from the inflammatory to the resolution phase of the macrophage response.

### Functional annotation of TTP binding sites during the inflammatory response

Our datasets include three pillars required for establishing a functionally annotated atlas of elements regulating post-transcriptionally the dynamics of inflammatory response of macrophages in the context of TTP function: (i) TTP binding sites, (ii) differential decay rates in WT versus ΔM macrophages, and (iii) differential expression data in WT versus ΔM macrophages. To navigate the atlas, we

designed a web interface publically available at http://ttp-atlas. univie.ac.at. The interface visualizes all three pillars in an integrated way and displays them for annotated genes expressed in murine macrophages. The atlas links the position and extent of TTP binding with the effects of TTP on the stability and abundance of the corresponding mRNA at the decisive steps of the inflammatory response.

Although our correlation analysis showed that TTP-dependent mRNA decay has a strong impact on the gene expression profile at the transition from inflammatory to resolution phase of the macrophage response (Fig 7D), it remained unclear whether this effect was a direct consequence of TTP binding to target mRNAs. Such effect might be also caused by indirect mechanisms including mRNA destabilization by factors which are themselves controlled by TTP. To answer this question, we used the functionally annotated TTP binding atlas for two types of correlation analyses: TTP binding versus differential decay at 3 and 6 h of LPS stimulation and TTP binding versus differential gene expression at 3 and 6 h of LPS stimulation. This analysis was restricted to the 3′ UTR datasets, since only binding to these regions has shown regulatory effects. Furthermore, we normalized TTP binding site scores to the expression of target mRNAs (i.e. RNA-Seq data) to assess relative strength of TTP binding to different targets (see Materials and Methods for description) and to analyze the correlation between binding strength and differential decay (Fig 8A) as well as between binding strength and differential expression (Fig 8B). The analyses of Pearson's coefficients showed that normalized TTP binding score (equation 1 in Materials and Methods) is correlated with both differential decay (Fig 8A) and differential expression (Fig 8B), an effect that becomes markedly stronger when switching from 3 to 6 h after LPS induction. The correlation with differential decay rates is even stronger than with differential expression, as judged by Pearson's coefficients.

Together, our datasets establish a functionally annotated atlas of TTP binding sites. The atlas together with the provided tools allows navigation and insights into regulation of the macrophage inflammatory response at the post-transcriptional level. Using the atlas, we provide evidence that the TTP-controlled transition of the macrophage response from the inflammatory into resolution phase of inflammation is a direct consequence of TTP binding to target mRNAs.

## Discussion

Our study represents a multilayered transcriptome-wide analysis of post-transcriptional gene regulation during the inflammatory response of the macrophage. The work integrates high-resolution mapping of TTP binding sites with analyses of TTP-dependent control of mRNA decay and abundance. Our approach provides a dynamic, rather than a snapshot, insight into the inflammatory response since the datasets include peak inflammatory as well as early resolution phases of the macrophage transcriptome. Comprehensive analyses of the datasets allowed us to establish a functionally annotated atlas of TTP binding sites for key phases of the macrophage response to an inflammatory stimulus. By multiple correlation assessments using the atlas data, we were able to show that TTP directly drives the conversion of the inflammatory phase into the resolution response.

One aim of this study was to identify at nucleotide resolution positions of TTP binding sites in immunostimulated macrophages

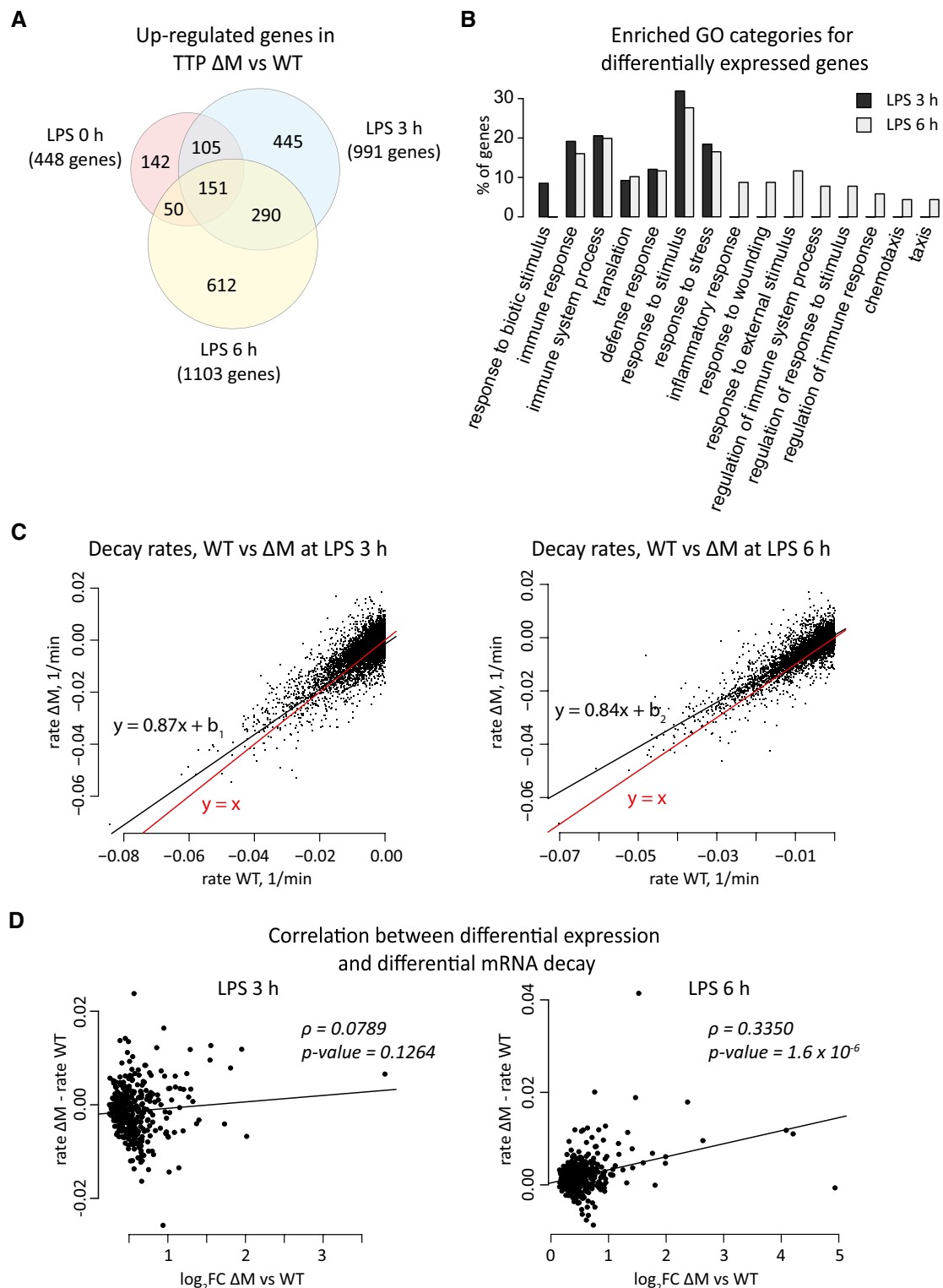

**Figure 7. TTP-dependent effect on mRNA decay and abundance increases when cells are entering the resolution phase of inflammation.**

A   The number of genes differentially expressed between WT and TTP-deficient (ΔM) BMDMs in untreated (0 h LPS) or LPS-treated BMDMs (3 h LPS or 6 h LPS).

B   Enriched GO biological process categories for genes differently expressed between WT and TTP-deficient BMDMs at 3 and 6 h of LPS treatment (enrichment FDR < 0.05).

C   Comparison of decay rates in WT and TTP-deficient (ΔM) BMDMs at 3 (left panel) and 6 h (right panel) of LPS treatment.

D   Correlation plots for differential expression and differential decay between WT and TTP-deficient (ΔM) BMDMs at 3 (left panel) and 6 h (right panel) of LPS treatment.

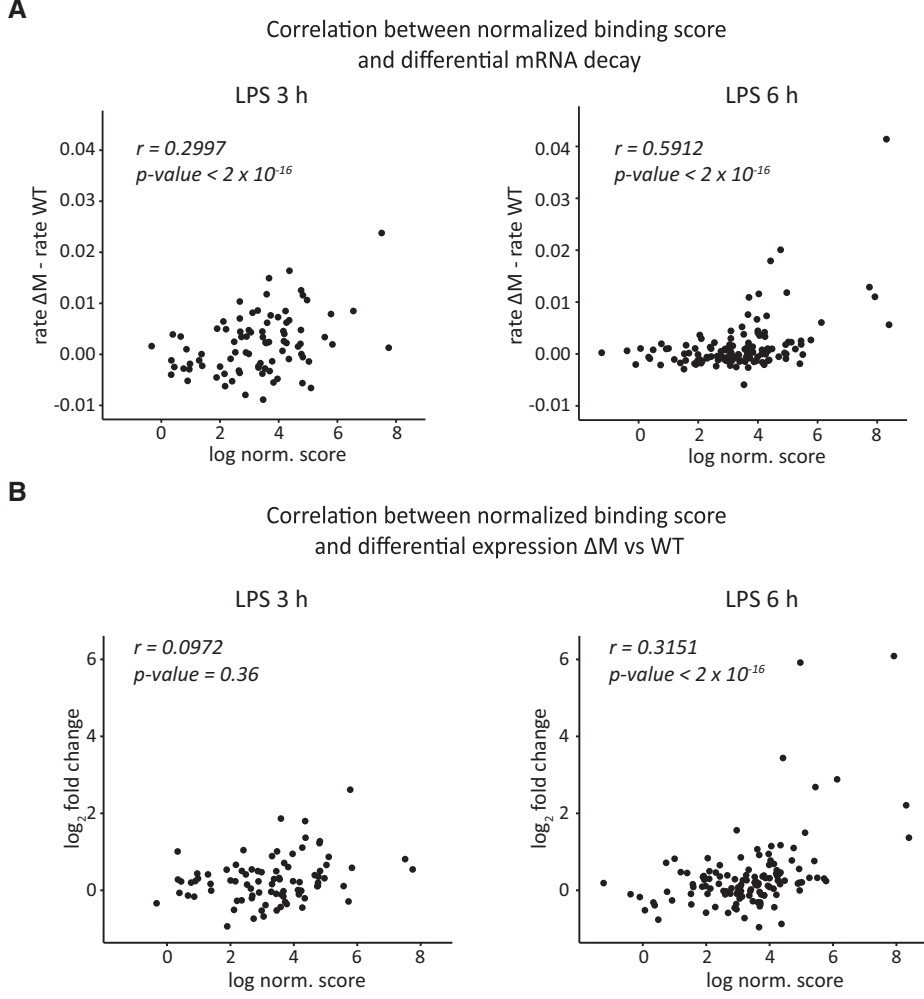

**Figure 8.  Pearson's correlation between normalized TTP binding versus differential RNA decay, and normalized TTP binding versus differential gene expression.**

A   Correlation between TTP binding and RNA decay increases from 3 h (Pearson's *r* = 0.30, 95% confidence intervals (CI) [0.094, 0.477], *P*-value < 2e-16, left panel) to 6 h (Pearson's *r* = 0.59, 95% CI [0.466, 0.693], *P*-value < 2e-16, right panel) after LPS induction.

B   Correlation between TTP binding and differential expression increases from 3 h (Pearson's *r* = 0.1, 95% CI [−0.113, 0.299], *P*-value 0.36, left panel) to 6 h (Pearson's *r* = 0.32, 95% CI [0.152, 0.462], *P*-value < 2e-16, right panel) after LPS induction.

in order to determine the impact of TTP binding on inflammatory gene expression. The selected experimental system is characterized by a high but controlled TTP expression, habitual regulation of TTP activity by phosphorylation and other post-translational modifications, and abundant presence of inflammation-associated physiological TTP target mRNAs (Brook *et al*, 2006; Hitti *et al*, 2006; Sandler & Stoecklin, 2008; Stoecklin *et al*, 2008; Schaljo *et al*, 2009; Kratochvill *et al*, 2011; Tiedje *et al*, 2012). Furthermore, macrophages and other myeloid cells represent so far the only cell types in which TTP expression has been linked with functional consequences *in vivo* (Carballo *et al*, 1998; Kratochvill *et al*, 2011; Qiu *et al*, 2012). Thus, our study examines TTP binding and function in a native and physiologically relevant environment. The very limited target overlap between our transcriptome-wide study and the reported study carried out in HEK293 cells overexpressing TTP (Mukherjee *et al*, 2014) is most likely caused by differences

in the experimental systems such as the missing expression of inflammation-associated TTP targets or different regulation of TTP activity and/or expression in HEK293 cells. In contrast to this study, most TTP targets known so far are present in our study, suggesting that coverage in our study is comprehensive. However, cell type-specific aspects of TTP binding cannot be excluded. Future studies should address this question by determining TTP binding sites in other primary cells, particularly cells of the hematopoietic system.

A minimum, albeit not sufficient, requirement for destabilization by TTP is binding to the 3′ UTR: Only TTP bound to the 3′ UTR, but not to other transcript regions, can destabilize the target. Nevertheless, most mRNAs targeted by TTP in their 3′ UTRs are stable, indicating that other factors contribute to destabilization by TTP. The extent of TTP occupancy at the target is not likely to be alone decisive for destabilization since TTP-destabilized mRNAs

often exhibit similar TTP binding site scores as stable transcripts (e.g. *Il6* versus *Sdc4*). The sequence of the TTP binding site does not appear to represent a critically important parameter since we detect the most preferred TTP binding sequence motif, the UAUUUAU heptamer, in stable as well as unstable TTP-bound mRNAs. These data implicate that the effects of TTP binding on target mRNA stability might be more context dependent than previously anticipated. For example, regions flanking the binding sites might contain other cis-acting elements such as seed sites for microRNAs. TTP has been reported to interact with miR16 to destabilize *Tnf* mRNA in HeLa cells (Qi *et al*, 2012), but a broader analysis, particularly in immune cells, is needed to assess the general role of microRNAs in TTP-dependent mRNA decay. Cis-acting elements might also exert their effects by recruiting other RNA-binding proteins which could act independently or in complexes with TTP to regulate mRNA decay. An interaction between TTP and the RNA-binding protein KSRP was reported, and this association enhanced the decay of the human *iNOS* mRNA (Linker *et al*, 2005). Evidence for a general contribution of KSRP to TTP-dependent mRNA decay has so far not been provided. TTP was also found to bind to several isoforms of the ARE-binding and mRNA-destabilizing factor AUF1, but the functional consequences of these interactions remain to be elucidated (Kedar *et al*, 2012).

We addressed the combinatorial effects of RNA-binding proteins on mRNA stability by genome-wide mapping of HuR binding sites in immunostimulated macrophages under the same conditions as applied to the TTP procedure. Our datasets establish that only a minor fraction of target mRNAs contain binding sites for both TTP and HuR in their 3′ UTRs. This finding excludes a direct competition of TTP and HuR for the same binding sites as a general means for the regulation of mRNA decay by these two proteins. Interestingly, the stability of mRNAs containing binding sites for both TTP and HuR is also, in general, not co-regulated by the two proteins although they can simultaneously bind. It remains to be elucidated whether such mRNAs are co-regulated by means of TTP-mediated mRNA degradation and HuR-facilitated translation, as reported for *Tnf* mRNA (Tiedje *et al*, 2012). The specific properties of binding sites targeted by both HuR and TTP, or conversely properties of binding sites targeted exclusively by either of the two proteins, need to be deciphered yet. However, our structure/sequence analysis of the binding site context revealed distinct properties for sites bound by TTP or HuR. Functional binding motifs for both TTP and HuR tend to be embedded in a larger AU-rich and weakly structured region, but for HuR, the lack of secondary structure seems to be more relevant than AU content. Despite the high incidence of intronic TTP binding sites found both in our study and by Mukherjee and colleagues (Mukherjee *et al*, 2014), the function of TTP binding to introns remains elusive. For *Irg1*, we find TTP to be associated with spliced-out introns, possibly to the lariat since the rate-limiting step of intron degradation is the cleavage of the lariat intermediate by the debranching enzyme DBR1 (Chapman & Boeke, 1991). TTP shuttles between the cytoplasm and the nucleus in a regulated manner (Johnson *et al*, 2002; Phillips *et al*, 2002; Stoecklin *et al*, 2004). Our data show that in contrast to cytoplasmic TTP, the levels of nuclear TTP remain stable throughout the inflammatory response. This

suggests that the interaction of TTP with introns might be less dynamic than with 3′ UTRs. Since our data do not indicate a role of intronic TTP binding in pre-mRNA processing, future studies should explore other options such as a role of TTP in the fate of intronic RNA.

A hallmark of activated macrophages is a highly dynamic and temporally coordinated expression of inflammation-associated transcripts, including many cytokine and chemokine mRNAs. The levels of these mRNAs are significantly regulated by precisely controlled mRNA decay such that they are allowed to accumulate during the onset of immune response but are degraded during resolution of inflammation. Although the mRNA-destabilizing activity of TTP was shown to increase during the inflammatory response, it remained unclear to what extent such regulation can determine the dynamics of the inflammatory gene expression profile (Kratochvill *et al*, 2011). Our current integrated approach shows a strong impact of TTP-dependent mRNA degradation on the transcriptome during the early resolution phase, whereas the onset of inflammation is only marginally influenced. Moreover, we show that the ability of TTP to dynamically regulate the inflammatory expression profile is a direct consequence of TTP binding to target mRNAs. The low impact of TTP-dependent decay on the gene expression profile during the early inflammatory phase contrasts with the substantial differences between the transcriptomes of WT and TTP-deficient macrophages in this phase. This surprising finding indicates that TTP directly controls the expression of only a few key drivers of the inflammatory response in the onset phase of inflammation. One such driver might be Tnf, which is strongly controlled by TTP already in the early response and can act as an autocrine amplifier by activating the central transcriptional inducer of cytokines NF-kappa-B (Hayden & Ghosh, 2012). The importance of the ability of TTP to directly control the macrophage response at the transition to the resolution phase is underpinned by the spectrum of TTP targets in this phase. These targets are significantly enriched in mRNAs coding for proteins involved in chemotaxis and cell migration, which are key processes in the establishment of non-resolving inflammation (Griffith *et al*, 2014). Thus, by interfering with these processes, TTP helps prevent chronic inflammation. This finding proposes a mechanistic explanation for the phenotype of animals lacking TTP expression in macrophages: These animals fail to resolve an inflammatory challenge but they do not spontaneously initiate inflammation and remain healthy under normal conditions (Kratochvill *et al*, 2011; Qiu *et al*, 2012).

Our study employs molecular systems biology approaches to integrate transcriptome-wide data on TTP binding sites with TTP-controlled mRNA stability and abundance. The study provides comprehensive insights into the regulation of the inflammatory transcriptome at the post-transcriptional level. It provides evidence that the function of RNA-binding proteins is more context dependent than previously anticipated. Cis- and trans-acting combinatorial effects similar to those known from transcription regulation will likely apply also to the regulation of mRNA decay. Using our integrative network, we identify a TTP-dependent switch, which drives the macrophage response from the inflammatory to the resolution phase. This finding improves our understanding of the molecular basis of chronic inflammatory diseases with important implications for the development of new therapeutical options.

# Materials and Methods

### Cell culture

Murine bone marrow-derived macrophages were produced from the bone marrow isolated from femur and tibia of 7- to 9-week-old TTPΔM mice or WT littermates. Macrophages were cultivated in DMEM (Sigma) supplemented with 10% FBS (Sigma), 100 U/ml penicillin (Sigma), 100 μg/ml streptomycin (Sigma), and CSF-1 derived from L929-cells as previously described (Kovarik et al, 1998). Mice for bone marrow isolation were bred and kept under specific pathogen-free conditions according to the recommendations of the Federation of European Laboratory Animal Science Association and were all on C57BL/6 background. All experiments involving bone marrow isolation were discussed with the institutional ethics committee and performed in accordance with the Austrian law for animal experiments (BGBl. I Nr. 114/2012) and in accordance with the guidelines recommended by the German Society of Laboratory Animals (GV-SOLAS). Animal experimental protocols were approved and authorized through the permission BMWF-66.006/0006-II/3b/2013 issued by the Austrian Ministry of Science to PK.

### PAR-iCLIP

UV crosslinking, immunoprecipitation, and library preparation were performed as described (Hafner et al, 2010; Konig et al, 2010). Briefly, BMDMs were cultured on 15-cm cell culture-treated dishes ($15 \times 10^6$ cells per plate) in medium supplemented with 100 μM 4-thiouridine (Sigma) for 16 h prior to experiment. Cells were then stimulated with LPS (10 ng/ml, Sigma) for 6 h, followed by washing with PBS (Sigma) and exposure to 365-nm UV light at 0.15 J/cm$^2$. Subsequently, $10^8$ cells were harvested in three volumes of lysis buffer [50 mM Tris–HCl pH 7.4 (Life Technologies), 100 mM NaCl (AppliChem), 1% NP-40 (AppliChem), 0.1% SDS (AppliChem), 0.5% sodium deoxycholate (AppliChem), protease inhibitor cocktail (Roche)]. Lysate was treated with 10 U/μl (2 U/μl, 0.5 U/μl) RNase I (Ambion) and 10 U DNase I (Roche) for 3 min at 37°C. TTP was immunoprecipitated with 50 μl TTP rabbit antiserum (Kratochvill et al, 2011) bound to Protein G Dynabeads (Invitrogen). After washing, $^{32}$P-labeled RNA adaptor (1 μM/sample) was ligated to 3′ end of RNA. Protein–RNA complexes were eluted from beads by incubating in 20 μl 1× NuPAGE LDS sample buffer (Life Technologies) for 10 min at 70°C, separated on pre-cast 4–12% NuPAGE Bis-Tris gel (Life Technologies), and transferred to nitrocellulose membrane using wet transfer (Bio-Rad). Radioactive signal was visualized using a phosphorimager screen, and bands corresponding to TTP–RNA complexes were cut out. RNA was recovered from the membrane using 10 μl proteinase K digestion (Roche) at 37°C for 40 min with shaking. RNA was reverse-transcribed with Super Script III (Life Technologies) and primers specific for 3′ adaptor. cDNA was fractionated using non-denaturing 6% PAAG gel (Life Technologies), and fragments in the ranges of 120–200 nt (high), 85–120 nt (medium), and 70–85 nt (low) were recovered by gel elution in TE buffer for 2 h at 37°C with intensive shaking followed by circularization using CircLigase II (Epicenter). Oligodeoxyribonucleotide complementary to BamHI site was annealed and digested with BamHI (Thermo Scientific) in order to linearize cDNA. cDNA was PCR-amplified with primers specific to linker regions. Library was subjected to high-throughput sequencing 100-bp single end on an Illumina HiSeq 2000 platform at the CSF NGS unit (http://csf.ac.at).

### RNA-Seq as control for PAR-iCLIP-Seq

Control RNA-Seq library was prepared using the same strategy as for the PAR-iCLIP library. Total RNA from $10^7$ BMDMs was isolated with TRIzol reagent (Life Technologies) according to the manufacturer's protocol. Ribosomal RNA was depleted using Ribo-Zero rRNA removal kit (Epicenter), and the remaining RNA was fragmented using RNA fragmentation reagent (Ambion). Enzymatically 5′ pre-adenylated RNA adaptor was ligated to RNA 3′ end. cDNA was produced using primer specific to 3′ adaptor sequence. cDNA was size-selected (200–300 nt) using non-denaturing gel, recovered, and circularized using CircLigase II (Epicenter). DNA oligo complementary to BamHI site was annealed and digested with BamHI (Thermo Scientific) in order to linearize cDNA. cDNA was then PCR-amplified with primers specific for linker regions. Library was subjected to high-throughput sequencing 100-bp paired end on an Illumina HiSeq platform at the CSF NGS unit (http://csf.ac.at).

### PAR-iCLIP and RNA-Seq data analysis

PAR-iCLIP and RNA-Seq reads were pre-processed (includes demultiplexing, barcode trimming, and adaptor removal with Cutadapt) (Martin, 2011). After quality control with FASTQC (http://www.bioinformatics.babraham.ac.uk/projects/fastqc/), reads were mapped to the mouse genome assembly mm9/NCBI37 using Segemehl v0.1.3-349M (Hoffmann et al, 2009). Uniquely mapped reads were extracted for further analysis. The replicates of TTP PAR-iCLIP contained 17349756, 16587192, and 16313702 mapped reads. The replicates of HuR PAR-iCLIP contained 38670127 and 38212687 mapped reads.

### PAR-iCLIP binding site finding and filtering

A binding site is defined as a region with a significantly higher number of read pileup at a given genomic position than would be expected by chance. We used the Pyicos modFDR method (Althammer et al, 2011) for binding site finding, together with a modified filtering algorithm for the use with PAR-iCLIP crosslink sites, which can be seen as reads of length one. Due to the nucleotide resolution of PAR-iCLIP, binding site width can range from one nucleotide for very sharp signals to several hundred nucleotides for regions with, for example, multiple consecutive binding sites. Our custom filtering method splits binding site regions surrounding the highest binding site signal, henceforth named summit in accordance with Pyicos, once certain height thresholds are reached. Cutoffs were defined based on signals detected in known TTP targets. Binding sites with a summit signal below 100 pileups are considered background and discarded. With a sliding window approach, starting from the summit, a binding site is first split when its height falls below 30% of the summit signal. Emerging subsites with a summit above this cutoff and 100 pileups are then recursively split when their signal falls below 10% of their

summit. Final split sites contain a high amount of crosslink signal and allow us to analyze protein binding sites with high resolution. Replicates of each experimental setup were analyzed separately. Width and position of binding sites vary slightly between experiments. For the ranked lists of TTP and HuR target genes, we collect binding sites from all replicates and filter out sites that do not have an overlap with sites in all other replicates. Resulting filtered sites were then applied to downstream analysis, for example, annotation and motif analysis.

### Binding site annotation

Crosslinks derived from uniquely mapped reads in binding site regions were annotated using the ENSEMBL Perl API for mouse annotation version 67. For gene statistics, we applied an exon first approach, where all transcript isoforms of a target gene are taken into account: A binding site region is classified as exonic if it occurs in an exon of at least one transcript isoform; it is intronic if it occurs in an intron of at least one isoform and never in an exon.

### Motif finding

The command line version of MEME (Bailey & Elkan, 1994) was used to detect overrepresented sequence motifs in the binding site. MEME builds its background model based on nucleotide frequencies of the input sequences. Since we aimed to identify motifs that are enriched in genomic elements (introns and 3′ UTRs) rather than all regions of PAR-iCLIP signal, we generated individual background models for those genomic elements found in ENSEMBL protein-coding genes. Binding site regions shorter than MEME's minimum sequence length (8 nt) were extended on both ends to a minimal length of 9 nt. Using the custom background models and the "any number of repeats" mode of MEME with motif length between 5 and 7 yielded the best results regarding both motif information and gene coverage.

### Quantification and normalization of PAR-iCLIP data

PAR-iCLIP cannot distinguish between poor binding sites in highly expressed targets and good binding sites in targets with low expression. In order to introduce a measure for binding site strength, we performed control RNA-Seq experiments using the same library preparation strategy as for PAR-iCLIP. Expression rates were calculated using Cufflinks v.2.0.2 [1] with ENSEMBL exon annotation as regions of interest. Gene-wide PAR-iCLIP signal was normalized by gene expression rates (FPKM) to define normalized binding (equation 1). Gene expression (equation 1. $FPKM_{gene}$) was calculated as the sum of FPKMs of all transcript isoforms ($FPKM_{transcript}$) of each gene containing the binding site within the mature mRNA. Only transcripts of FPKM $\geq$ 10 were considered, as we expect TTP targets under inflammatory stress to be strongly expressed.

$$\text{Score}_{normalized} = \frac{\text{Binding site area}}{FPKM_{gene} + (\text{median FPKM} * \alpha)} \qquad (1)$$

Sparse data correction (median FPKM transcript was added to FPKM gene before normalization) was applied, to avoid spurious high GeneScores of very low expressed genes.

### ROC analysis

To investigate the sequence and structure properties of bound and unbound RNA sequences, we derived coordinates of AREs of type WAUUUAW and UUUKUUU from AREsite (Gruber *et al*, 2011). Those AREs resemble the most over-represented motifs in TTP (WAUUUAW) and HuR (UUUKUUU) binding sites we identified in this study. Intersection with our dataset allowed us to create a "positive" (bound) and "negative" (unbound) set of sequence motifs, which was used to train linear discriminators based on sequence (AU content) and structuredness. The latter is defined by the accessibility of a region, which means the probability to find paired nucleotides. We calculated structuredness with RNAplfold (Bernhart *et al*, 2011), in terms of opening energy.

### Comparison of datasets for TTP binding in BMDMs and HEK293 cells

In order to identify homologous TTP binding sites in humans and mouse, we extracted syntenic regions using the liftover tool of the UCSC binary collection (Kent *et al*, 2002). Coordinates (human genome assembly hg19) of previously identified TTP binding sites in human HEK293 cells (Mukherjee *et al*, 2014) were lifted to mouse coordinates (NCBI m37 [mm9], ENSEMBL annotation 67) using -minMatch = 0.05. Overlapping regions for further analysis were extracted using the BEDTools suite (Quinlan & Hall, 2010).

### Library preparation for differential gene expression and mRNA decay analysis

BMDMs were stimulated with LPS (10 ng/ml) for 3, 6 h or left untreated. Medium was then replaced by a fresh medium containing actinomycin D (5 µg/ml). Total RNA was isolated using TRIzol reagent 0, 45, 90 min after the addition of actinomycin D. RNA-Seq libraries were prepared using SENSE mRNA-Seq Library Prep Kit (Lexogen). Libraries were subjected to 100-bp single-end high-throughput sequencing on an Illumina HiSeq 2500 platform at the CSF NGS unit (http://csf.ac.at). For each time point, three biological replicates were used.

### Differential gene expression and mRNA decay analysis

RNA-Seq reads were pre-processed (including demultiplexing, barcode, adaptor and quality trimming, quality control using FASTQC, http://www.bioinformatics.babraham.ac.uk/projects/fastqc/). The remaining reads were mapped to the mouse genome assembly GRCm38/mm10 using gsnap (Wu & Watanabe, 2005). Quality control of mapping was performed using RNA-SeQC (DeLuca *et al*, 2012). Transcripts were quantified using the Mix$^2$ RNA-Seq data analysis software (Lexogen). Differential expression analysis was performed based on read counts (FPKM) in gene models using DESeq2 (Love *et al*, 2014). Decay of mRNA was quantified assuming a model of exponential decay (Ross, 1995). For each gene, a linear model was fitted with log2-transformed FPKM as target variables and time after actinomycin D addition as explanatory variable. Therefore, the slope of the linear model estimates the rate of mRNA decay for each gene. Half-life may be calculated as the inverse of the rate. We note that the estimates of mRNA decay at

the different time points, that is, 3 and 6 h of LPS treatment, are from independent replicates and thus not trivially correlated. Pearson's correlations between various variables and estimates were calculated as descriptive statistics.

## Quantification of gene expression by qRT–PCR

Total RNA was isolated using TRIzol reagent (Life Technologies) according to the manufacturer's protocol. DNase digestion was performed using recombinant DNase I (Roche). RNA was then reverse-transcribed by SuperScript III reverse transcriptase (Life Technologies). cDNA was quantified using HOT FIREPol EvaGreen qPCR Supermix (Solis BioDyne) on realplex Mastercycler (Eppendorf) as described before (Morrison *et al*, 1998). mRNA expression of the housekeeping gene *Hprt* was used for normalization.

## Measurement of mRNA stability by qRT–PCR

BMDMs ($2 \times 10^6$ cells) were stimulated with LPS (10 ng/ml) as described in figure legends. Medium was then replaced by a fresh medium containing actinomycin D (5 µg/ml). Total RNA was isolated using TRIzol reagent 0, 30, 60 min after the addition of actinomycin D. Remaining mRNA was quantified using qRT–PCR as described above.

## RNA immunoprecipitation (RNA-IP)

BMDMs were cultured in the medium supplemented with 100 µM 4-thiouridine (Sigma) for 16 h prior to the experiment. BMDMs were stimulated with LPS (10 ng/ml) for 6 h. Cells were washed with PBS and crosslinked using 365-nm UV light at 0.15 J/cm². Subsequently, the cells were harvested in three volumes of lysis buffer [50 mM Tris–HCl pH 7.4, 100 mM NaCl, 1% NP-40, 0.1% SDS, 0.5% sodium deoxycholate, protease inhibitor cocktail (Roche)]. Lysate was treated with DNase I (Roche) for 5 min at 37°C. TTP-bound RNA was immunoprecipitated using 50 µl TTP rabbit antiserum (Kratochvill *et al*, 2011) bound to Protein G Dynabeads (Invitrogen) for 3 h at 4°C. Beads were washed in lysis buffer followed by wash buffer (20 mM Tris–HCl pH 7.4, 10 mM MgCl₂, 0.2% Tween 20). TTP–RNA complexes were eluted from beads at 70°C for 15 min in 100 µl elution buffer (50 mM Tris–HCl pH 6.5, 100 mM DTT, 2% SDS). RNA was recovered using proteinase K digestion and the subsequent phenol–chloroform extraction and ethanol precipitation. cDNA was produced using random octamer primers. Precipitated RNA was quantified using qRT–PCR as described above. Relative mRNA levels were calculated as $2^{25-Ct}$.

## Native RNA immunoprecipitation (native RNA-IP)

BMDMs ($20 \times 10^6$ cell) were stimulated with LPS (10 ng/ml) for 3 or 6 h. Cells were washed with ice-cold PBS and subsequently harvested in native lysis buffer [10 mM Tris–HCl pH 7.4, 30 mM tetrasodiumpyrophosphate, 50 mM NaCl, 50 mM NaF, 1% Triton X-100, protease inhibitor cocktail (Roche), RNase inhibitor]. Lysate was treated with DNase I (Roche) for 5 min at 37°C. TTP-bound RNA was immunoprecipitated using 50 µl TTP rabbit antiserum (Kratochvill *et al*, 2011) bound to Protein G Dynabeads (Invitrogen) for 3 h at 4°C. Beads were washed three times in lysis buffer.

TTP–RNA complexes were eluted from beads at 70°C for 15 min in 50 µl elution buffer (50 mM Tris–HCl pH 6.5, 100 mM DTT, 2% SDS). RNA was recovered using phenol–chloroform extraction and ethanol precipitation. cDNA was produced using mixture of random octamer and oligo(dT) primers. Precipitated RNA was quantified using qRT–PCR as described above. Relative mRNA levels were calculated as $2^{25-Ct}$.

## Sequential RNA immunoprecipitation

BMDMs ($20 \times 10^6$ cells) were stimulated with LPS (10 ng/ml) for 6 h. Cells were washed with ice-cold PBS and subsequently harvested in native lysis buffer (10 mM Tris–HCl pH 7.4, 30 mM tetrasodiumpyrophosphate, 50 mM NaCl, 50 mM NaF, 1% Triton X-100, protease inhibitor cocktail (Roche), RNase inhibitor). TTP-bound RNA was immunoprecipitated using 30 µl TTP rabbit antiserum (Kratochvill *et al*, 2011) bound to Protein G Dynabeads (Invitrogen) for 16 h at 4°C. Beads were washed three times in lysis buffer. TTP–RNA complexes were eluted using a TTP peptide in lysis buffer at 37°C for 30 min under agitation. Eluate was immunoprecipitated using HuR antibody (clone 3A2, Thermo Fisher Scientific) or tubulin antibodies (clone DM1A Sigma) as negative control bound to Protein G Dynabeads (Invitrogen) for 16 h at 4°C. Beads were washed three times in lysis buffer. TTP–HuR–RNA complexes were eluted from beads at 70°C for 15 min in 50 µl elution buffer (50 mM Tris–HCl pH 6.5, 100 mM DTT, 2% SDS). RNA was recovered using phenol–chloroform extraction and ethanol precipitation. cDNA was produced using mixture of random octamer and oligo(dT) primers and subsequently PCR-amplified using specific primers. PCR products were analyzed by agarose electrophoresis.

## Nuclear/cytosolic fractionation and immunoblotting

Procedure for the separation of cytosolic from nuclear fractions was a modification of a reported protocol (Sadzak *et al*, 2008). BMDMs ($2 \times 10^7$ cells) were washed in PBS (Sigma), scraped in PBS, and harvested by centrifugation. Cytoplasmic fraction was extracted using cytoplasmic lysis buffer [10 mM HEPES pH 7.5, 60 mM KCl, 1 mM EDTA, 1 mM DTT, 0.1% NP-40, protease inhibitor cocktail (Roche)]. Nuclei were washed twice in wash buffer [10 mM HEPES pH 7.5, 60 mM KCl, 1 mM EDTA, 1 mM DTT, protease inhibitor cocktail (Roche)]. Nuclei were lysed in 100 µl 1× SDS loading buffer and sonicated to reduce sample viscosity. Sample preparation, SDS–PAGE, and immunoblotting were performed as described (Sadzak *et al*, 2008).

## Statistical analysis and visualization

Exploratory data analysis, visualization, and statistical testing were performed with R-project (http://www.R-project.org) or perl scripts using unpaired two-sample unequal variances *t*-test (Welsh's *t*-test), Mann–Whitney test, Kolmogorov–Smirnov test, linear regression model, Pearson's correlation test as indicated in figure legends. Venn diagrams were produced using Venn Diagram Plotter (Pan-Omics Research). RNA-Seq read coverage was calculated with coverageBed from BEDTools suite (Quinlan & Hall, 2010).

## GO enrichment analysis

GO enrichment analysis was performed using DAVID version 6.7 (Huang da *et al*, 2009a,b). Ensembl IDs for genes corresponding to transcripts targeted by TTP or HuR in 3′ UTR were submitted to DAVID Functional Annotation Tool (http://david.abcc.ncifcrf.gov/).

## Data availability

The raw as well as processed binding and expression data are accessible via GEO under reference series number GSE63468 (http://www.ncbi.nlm.nih.gov/geo/query/acc.cgi?acc = GSE63468).

## Primer list

Murine Hprt fw GCAGTCCCAGCGTCGTGAT
Murine Hprt rv CGAGCAAGTCTTTCAGTCCTGTC
Murine Tnf fw GATCGGTCCCCAAAGGGATG
Murine Tnf rv CACTTGGTGGTTTGCTACGAC
Murine Cxcl2 fw GCCCAGACAGAAGTCATAG
Murine Cxcl2 rv GTCAGTTAGCCTTGCCTT
Murine Tapbp fw ATACTTCAAGGTGGATGACCCG
Murine Tapbp rv CTGCTCCGGACTCAGACTTC
Murine Gnb1 fw GAACTAAAGCCAGGAGCAGCAG
Murine Gnb1 rv TGTGGGTGTCCACGTTAGTATG
Murine Ccl3 fw TCTCCTACAGCCGGAAGATCCC
Murine Ccl3 rv CATTCAGTTCCAGGTCAGTGATG
Murine Zfp36 fw CTCTGCCATCTACGAGAGCC
Murine Zfp36 rv GATGGAGTCCGAGTTTATGTTCC
Murine Fos fw GGAGAATCCGAAGGGAACGG
Murine Fos rv CTGTCTCCGCTTGGAGTGTAT
Murine Ccl12 fw GAAGCTGTGATCTTCAGGACC
Murine Ccl12 rv TCTTAACCCACTTCTCCTTGGG
Murine Cxcl1 fw TGCACCCAAACCGAAGTCATAG
Murine Cxcl1 rv TTGTATAGTGTTGTCAGAAGCCAGC
Murine Il10 fw GGACTTTAAGGGTTACTTGGGTTGCC
Murine Il10 rv CATGTATGCTTCTATGCAGTTGATGA
Murine Irf1 fw CCGAAGACCTTATGAAGCTCTTTG
Murine Irf1 rv GCAAGTATCCCTTGCCATCG
Murine Irg1 fw AGTTTTCTGGCCTCGACCTG
Murine Irg1 rv AGAGGGAGGGTGGAATCTCT
Murine Irg1 intron 3 fw TCAGAAATCTGAGGTGGCCTG
Murine Irg1 intron 3 rv CAGATACTGGGAGCCACAACA
Murine Irg1 intron 4 fw AACTTTCCAGCCCCACTAGC
Murine Irg1 intron 4 rv TAGGAACAGGCCACTGGGTA
Murine Irg1 exon 4–intron 4 fw CACAGCTCTATCGGAAGCCC
Murine Irg1 exon 4–intron 4 rv CCAGCCTCTAAGCCAGACAG

**Expanded View** for this article is available online.

## Acknowledgements

We thank Meghan Lybecker and Birgit Strobl for critical reading of the manuscript and Nadia Sedlyarova for valuable comments. We are grateful to Heinz Ekker from the NGS facility at CSF for help with sequencing data pre-processing. This work was supported by the University of Vienna Research platform 323500 to PK and IH, by a research cluster initiative supported jointly by the Medical University of Vienna and the University of Vienna to PK, by the Austrian Science Fund (FWF) Grant SFB 43 to PK and IH, and by funding from the European Union Seventh Framework Programme Marie Curie Initial Training Networks (FP7-PEOPLE-2012-ITN) for the project INBIONET under grant agreement PITN-GA-2012-316682.

## Author contributions

VS, JF, IH, and PK perceived the idea of the study and designed the experiments. VS, JF, and FE performed the experiments and data analysis. JH, LS, MI, KK, AT, and CV contributed to parts of the study. VS and FE designed web interface. VS programmed web interface. VS, JF, IH, and PK wrote the paper.

## Conflict of interest

The authors declare that they have no conflict of interest.

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
