## [Review Process File · Molecular Systems Biology]

Tristetraprolin binding site atlas in the macrophage transcriptome reveals a switch for inflammation resolution

Vitaly Sedlyarov, Jörg Fallmann, Florian Ebner, Jakob Huemer, Lucy Sneezum, Masa Ivin, Kristina Kreiner, Andrea Tanzer, Claus Vogl, Ivo Hofacker and Pavel Kovarik

Corresponding author: Ivo Hofacker & Pavel Kovarik, University of Vienna

Review timeline:	Submission date:	09 October 2015
	Editorial Decision:	24 November 2015
	Revision received:	24 February 2016
	Editorial Decision:	01 April 2016
	Revision received:	07 April 2016
	Accepted:	13 April 2016

Editor: Maria Polychronidou

Transaction Report:

1st Editorial Decision

24 November 2015

Thank you again for submitting your work to Molecular Systems Biology. We have now heard back from the three referees who agreed to evaluate your manuscript. Overall, the reviewers appreciate that the presented analyses are potentially interesting since they are performed in a physiologically relevant context. However, they raise a series of concerns, which should be carefully addressed in a revision of the manuscript.

Without repeating all the comments listed below, some of the more fundamental issues are the following:

- As reviewers #1 and #3 point out, it should be clarified whether several replicates were used in the presented analyses and to provide the appropriate analyses/comparisons of all biological replicates.

- Additional analyses are required to show that TTP has a direct effect on mRNA stability and to provide further insights into the role of TTP binding to stable mRNAs and to spliced-out introns without affecting RNA processing (currently shown only for the Irg1 transcript).

- Reviewer #2 is not convinced that CLIP-seq is a suitable technique for analyzing TTP binding dynamics. We have circulated the reports to all reviewers as part of our 'pre-decision cross-commenting' policy. During this process, reviewer #3, mentioned: "I partially agree with Reviewer #2. Although crosslinking is not complete (we have measured that crosslinking efficiency goes from 1 to 40% of the total protein in a protein-dependent basis) if the authors have controlled the experiment properly using three replicates and can show that the differences between time points are superior to what one would expect by chance (technical and biological variation), I think the conclusions are acceptable (as I described in my report). However, I would suggest to tone down the

conclusions regarding this part. What I would like to see is an experimental validation that the differential TTP binding to these transcripts is indeed occurring (and what is observed is not just experimental variation). This can be easily carried out by immunoprecipitating the protein in absence of RNase treatment and measuring the dynamically bound mRNAs by RT-qPCR. This would provide a strong argument (not dependent on crosslinking efficiency) about the dynamical behavior of TTP at the two different time points. The authors already did something similar to test the intron-binding properties of TTP. Obviously there is not (yet) an optimal tool to look at dynamic RNA binding of RBPs in two different conditions. However, I think this is a nice approximation and if the results are validated, I would be satisfied."

- Moreover, Reviewer #3 refers to the need to carefully edit the text in order to make the main findings and the related analyses more easily accessible to a broad audience.

REFeree REPORTS

Reviewer #1:

In this manuscript Sedlyarov and colleagues examine the TTP-dependent expression changes of the macrophage transcriptome in both, inflammatory and resolution phases as a response to LPS stimulation. The authors use PAR-iCLIP of TTP, gene expression profiling and mRNA decay studies to generate an atlas of regulation of the macrophage inflammatory response at the posttranscriptional level. Sedlyarov and coworkers provide evidence that the TTP-controlled transition of the macrophage response from the inflammatory into resolution phase of inflammation is a direct consequence of TTP binding to target mRNAs.

This manuscript is one of the rare studies that examines the function of an RNA-binding protein in its proper biological context. Though TTP has been analyzed before by similar high throughput approaches by Mukherjee and colleagues (2014) with similar functional findings, this manuscript examines the effect of TTP on gene expression in the macrophage inflammatory response. The work is in general well conducted and of interest to a broader audience. Before considering this manuscript for publication, some of the following issues should be resolved.

Major:

The authors state in that "one replicate was used for further detailed analysis". Is this statement restricted to the analysis of the binding site distribution (UTRs, CDS and intron) or is this approach applied elsewhere. The reviewer raises this question since in the Methods section the authors mention that "for the ranked lists of TTP and HuR target genes, we collect binding sites from all replicates and filter out sites that do not have an overlap with sites in all other replicates. Resulting filtered sites were then applied to downstream analysis, e.g. annotation and motif analysis." Is Dataset EV1 generated by the filtering approach?

In this context the reviewer strongly suggests to build a consensus binding site set that consists of at least 2 of the three CLIP libraries contributing to a given binding site. The consensus set should be used for all analyses.

The authors use a RIP-qRT/PCR approach to show that TTP binds to the spliced out intron of the Irg1 transcript and to confirm the PAR-iCLIP results. Since this was only shown for Irg1 the authors should rephrase their strong conclusion "TTP molecules targeting intronic sequences are predominantly associated with spliced-out introns. Binding of TTP to introns does not affect processing and splicing of the targeted transcript. "

The authors should provide a comparison of the two biological CLIP replicates of HuR. It is unclear whether a HuR consensus binding set was used for the analysis or a single CLIP data set (see above).

The authors PAR-iCLIP datasets suggests that most mRNAs are not co-regulated by HuR and TTP since only a small fraction of genes are targeted by both proteins. How does mRNA decay change in this small subset of coregulated transcripts when TTP is depleted. Are they stabilized by HuR?

The authors further examine mRNA destabilization during the inflammatory response. To address

this question the authors mapped TTP binding sites at 3 h of macrophage stimulation with LPS by using the same approach as for the 6 h analysis described above. Was it indeed the same approach with three replicates? How many TTP CLIP replicates were generated and how do they correlate? The authors showed that a strong impact of TTP on gene expression profiles in the peak inflammatory phase (3 h of LPS stimulation) and early resolution phase (6 h of LPS stimulation) of the macrophage response. The authors state that "such analysis includes both direct and indirect effects of TTP function. Next Sedlyarov and colleagues assess the direct effects of TTP-dependent mRNA destabilization by examining their correlation with gene expression profiles. The analysis showed an increasing influence of TTP-dependent mRNA decay on shaping the expression profile at the transition to the resolution phase of the inflammatory response. These data establish that TTP-mediated mRNA decay directly controls the switch from the inflammatory to the resolution phase of the macrophage response. Furthermore the correlation of normalized PAR-iCLIP signal with differential decay and differential expression (Figure 8A+B) showed a strong increase of Pearson correlation coefficient during progression of macrophage response from 3 h to 6 h after LPS induction suggesting a direct effect by TTP. The authors should confirm this meta-transcript analyses by qRT-PCR mRNA decay measurements on some selected TTP-bound transcripts to support the conclusion that this is a direct effect of TTP.

Minor:

- The authors state "the highest number of TTP binding sites were located in introns (64%) and 3' UTRs (35%). The number of binding sites should be normalized to the length of intron and 3UTRs.
- The authors state that "this group represents mRNAs which might be regulated by competition of HuR and TTP for binding, similar as reported for Tnf mRNA (Tiedje et al, 2012)." In this context the reference Mukherjee et al. Genome Biology (2014) should be included.

Reviewer #2:

This study by Kovarik and colleagues uses some advanced techniques to attempt quantification of combinatorial interactions between TTP and HuR, and to determine how TTP may affect different subsets of mRNAs dynamically following activation of mouse macrophage. They claim to define a "switch" that correlates with activation of the macrophage following treatment with LPS.

The major problem with this paper is that PAR-CLIP is not a quantitative procedure. This is widely known. PAR-CLIP like other CLIP methods is very useful for mapping binding sites of most RNA binding proteins. However, the efficiency of UV crosslinking is poor and no publication has shown CLIP is able to saturate binding sites; in fact when has been measured saturation could not be demonstrated. This fact affects most of the experiments in this study; unfortunately, the field does not yet have a single technology that can solve the combinatorial problem quantitatively. Of course, this can be done using traditional biochemistry and structural analysis of purified RNAs and RNA binding proteins. Therefore, CLIP-seq techniques are not suitable for global analysis of biological dynamics like macrophage activation. Indeed, the Mukherjee 2014 paper was not able to directly solve the combinatorial problem quantitatively. Moreover, PAR-CLIP has significant background that would have to be accounted for prior to any rigorous quantitative normalization. Thus, without binding site coverage of the entire transcriptome and consideration of background the authors cannot rigorously measure the degree of overlap between TTP and HuR when using PAR-CLIP. Indeed, the publications to date regarding overlaps between TTP and HuR have a very wide variation in binding site assignments, as one would expect when binding sites are not saturated.

Most conclusions in this manuscript are already claimed or demonstrated in other published studies. For example, many studies, including that of Hao and Baltimore have addressed changes in mRNAs encoding transcription factors and cytokines following macrophage activation. Many studies have shown that TTP controls TNF-alpha and other cytokine mRNAs but there is ambiguity in global reports regarding active versus silent targets of TTP, whether TTP mRNAs increase or decrease stability, etc. Overall, global TTP mRNA targeting experiments have shown mixed conclusions regarding what and how TTP targets subsets of mRNAs encoding cytokines and other immune regulatory factors, and this manuscript does not resolve the confusion in the field.

Reviewer #3:

Sedlyarov et al reports here a genome-wide approach that integrates PAR-iCLIP, mRNA expression profiles and RNA decay to understand the role of TPP posttranscriptional control of gene expression in inflammation. To this end, authors applied these sequencing approaches to BMDMs induced with LPS. They show that TTP binds to 3' UTRs and intronic sequences, but only promote RNA degradation (in few cases) when bound to the 3' UTR. They compared TTP RNA-binding profiles to HuR PAR-iCLIP data showing no interplay for most of the target mRNAs and potential competitive mechanisms between the two proteins for a small set. Finally, they compared the binding profile of TTP in acute or inflammation-resolving phase showing very interesting, but subtle differences. I must admit this work is technically impressive and would be of interest for RNA biologists and immunologists. The experiments (especially PAR-iCLIP) are very well controlled. I consider very positive that authors employ here a 1) primary cells of the immune system, 2) different LPS-treatment conditions and 3) comparative analysis of two RBPs with regulatory roles in inflammation. Although a previous manuscript reporting TTP binding profile in HEK293, I nevertheless think this work add new data generated in a more physiological context.

MAJOR COMMENTS

- 1) I must say the text is sometimes too technical and difficult to follow and I found myself reading a paragraph 3 or 4 times to fully understand what the authors wanted to say. In this regard, I missed a more didactic description of the results to guide scientists from other fields throughout the manuscript. Although I am familiar with PAR-iCLIP, a detailed description of the method will help to non-experts. I found especially hard the description of the results associated to figure 7 and 8, and I do believe that a more didactic approach would help to highlight the quality of this work.
- 2) One of the most interesting aspects is the binding to fairly stable introns. This invokes sponge-like mechanisms to regulate TTP function rather than the opposite. I wonder whether these introns are indeed circular. I understand that unveiling a potential regulatory mechanism represents a whole project by itself, but some preliminary data in this regard would be of high interest.
- 3) The difference between binding in the 3' UTR increases from 157 to 198. Is this difference superior to what one may expect from technical variance between replicates at each time point? If it is not an artefact, differential binding can be easily validated by crosslinking, IP and RT-qPCR of few mRNAs.
- 4) Following (1), It's a bit confusing that TTP only affect stability of a limited number of transcripts in a TTP-dependent manner, but in figure 7-8, this proteins shapes the transcriptome. A more diligent explanation of the data would help to assess correctly all these data.

MINOR POINTS

- a) What does it mean: "one replicate was used for further detailed analysis"? This doesn't sound right.
- ab Figure 1A and 6A could be potentially biased due to differences in sequence length. i.e. if introns are in average longer than ORFs is not surprising to find enrichment of introns upon unspecific binding. The same applies for 5' and 3' UTRs. Normalisation of the data to sequence length can be used to estimate whether the binding in these regions is higher than the expected by random chance.
- c) Authors mention that the PAR-CLIP determined sites correlate well in to the sites previously reported for Cxcl1, Il1a and Cxcl2. I wonder if it could be possible to represent these data in a schematic. It could be quite illustrative about how these system-wide data fit into previous mechanistic studies.
- d) When correlating the binding profile to secondary structure and primary sequence I wonder why the authors used 7 nt (sliding?) windows. Stems can be potentially form from longer distances and I am afraid this analysis could miss a number of secondary structures. Perhaps I understood wrong this part, if so, some clarification would help.

1st Revision - authors' response

24 February 2016

Point-by-point reply (cited parts are in *italic*):

Reviewer #1:

1) *"The authors state in that "one replicate was used for further detailed analysis". Is this statement restricted to the analysis of the binding site distribution (UTRs, CDS and intron) or is this approach*

applied elsewhere. The reviewer raises this question since in the Methods section the authors mention that "for the ranked lists of TTP and HuR target genes, we collect binding sites from all replicates and filter out sites that do not have an overlap with sites in all other replicates. Resulting filtered sites were then applied to downstream analysis, e.g. annotation and motif analysis." Is Dataset EV1 generated by the filtering approach?

In this context the reviewer strongly suggests to build a consensus binding site set that consists of at least 2 of the three CLIP libraries contributing to a given binding site. The consensus set should be used for all analyses."

We apologize for the inconsistent statements in the main text and the Methods part. We confirm that the statements in the original Methods part were correct: a consensus binding site dataset has been generated from all replicates, and this dataset has been used throughout the study for all analyses. The incorrect parts in the main text have been changed accordingly (Results section 1, paragraph 5).

2) "The authors use a RIP-qRT/PCR approach to show that TTP binds to the spliced out intron of the *Irg1* transcript and to confirm the PAR-iCLIP results. Since this was only shown for *Irg1* the authors should rephrase their strong conclusion "TTP molecules targeting intronic sequences are predominantly associated with spliced-out introns. Binding of TTP to introns does not affect processing and splicing of the targeted transcript."

We have modified the text as suggested by the reviewer. First, the title of the corresponding Results section (Results section 3) has been changed from "TTP targeting intronic RNA is associated with spliced-out introns but has no role in target RNA processing" into "TTP binding to introns does not interfere with processing of target RNA". Second, the general statement on TTP binding to spliced-out introns has been deleted from the concluding sentences of this Results section.

3) "The authors should provide a comparison of the two biological CLIP replicates of HuR. It is unclear whether a HuR consensus binding set was used for the analysis or a single CLIP data set (see above)."

We used the same procedure for the analysis of HuR CLIP data as for TTP CLIP (see point 1): The two HuR CLIP replicates, which we obtained, were used to generate a consensus binding site dataset. Only binding sites present in this dataset were subjected to further analyses. The text in the corresponding Results section has been amended accordingly (Results section 4, paragraph 1).

4) "The authors PAR-iCLIP datasets suggests that most mRNAs are not co-regulated by HuR and TTP since only a small fraction of genes are targeted by both proteins. How does mRNA decay change in this small subset of coregulated transcripts when TTP is depleted. Are they stabilized by HuR?"

As suggested by the reviewer, we have compared differential decay of mRNAs containing overlapping TTP and HuR sites with mRNAs containing only TTP sites (using Kolmogorov-Smirnov test) (**new Figure 4J, upper panel**). The analysis showed no significant differences between these two groups of mRNAs. A similar comparison of differential expression (**new Figure 4J, lower panel**) also did not reveal significant differences between these two mRNA groups. These analyses indicate that the stability of mRNAs targeted by both TTP and HuR is, in general, not differently regulated as compared to mRNAs targeted by TTP only. Thus, a co-regulation by TTP and HuR at the level of mRNA stability is not a general rule even in cases when these proteins target the same mRNA. It should be noted, that this analysis aimed at revealing common principles so that it is possible that a co-regulation of some mRNAs can occur.

As a novel piece of data supporting binding of TTP and HuR to the same targets we show sequential RNA-IPs (first IP: TTP RNA-IP; second IP: HuR RNA-IP, after elution of the TTP-RNA complexes from the beads using TTP peptide). The data (**new Figure 4H**) show that TTP and HuR can bind simultaneously to mRNAs containing binding sites for both proteins (e.g. *Tnf* and *Cxcl2* mRNAs). No HuR binding was detected to targets containing only HuR sites (e.g., *Tapbp* and *Gnb1* mRNAs). Together, our findings are in agreement with the study by Tiedje et al (2012) which reported co-regulation of *Tnf* mRNA by TTP and HuR at the level of mRNA translation rather than mRNA stability.

5) "The authors further examine mRNA destabilization during the inflammatory response. To address this question the authors mapped TTP binding sites at 3 h of macrophage stimulation with LPS by using the same approach as for the 6 h analysis described above. Was it indeed the same approach with three replicates? How many TTP CLIP replicates were generated and how do they correlate?"

The authors showed that a strong impact of TTP on gene expression profiles in the peak inflammatory phase (3 h of LPS stimulation) and early resolution phase (6 h of LPS stimulation) of the macrophage response. The authors state that "such analysis includes both direct and indirect effects of TTP function. Next Sedlyarov and colleagues assess the direct effects of TTP-dependent mRNA destabilization by examining their correlation with gene expression profiles. The analysis showed an increasing influence of TTP-dependent mRNA decay on shaping the expression profile at the transition to the resolution phase of the inflammatory response. These data establish that TTP-mediated mRNA decay directly controls the switch from the inflammatory to the resolution phase of the macrophage response. Furthermore the correlation of normalized PAR-iCLIP signal with differential decay and differential expression (Figure 8A+B) showed a strong increase of Pearson correlation coefficient during progression of macrophage response from 3 h to 6 h after LPS induction suggesting a direct effect by TTP. The authors should confirm this meta-transcript analyses by qRT-PCR mRNA decay measurements on some selected TTP-bound transcripts to support the conclusion that this is a direct effect of TTP."

The PAR-iCLIP analysis for the 3 h LPS time point was carried out using the same approach as described for the 6 h LPS treatment (see reply to point 1 above): three replicates were used for the generation of a consensus binding site dataset. As suggested by the reviewer, we have validated the metadata using two approaches. First, we determined mRNA decay rates for several targets using qRT-PCR. The data (**new Figure EV5**) confirm that *Cxcl1*, *Ccl4*, *Il10* and *Zfp3612* mRNAs were more strongly destabilized by TTP at 6 h than 3 h of LPS treatment whereas *Irf1* mRNA was decaying similarly at both time points, consistent with the Dataset EV2. Second, we validated the increased TTP binding at 6 h of LPS stimulation using native RNA-IP (**new Figure EV4**): TTP binding to *Fos* and *Ccl12* mRNAs strongly increased between 3 and 6 h of LPS treatment whereas binding to *Tnf*, *Ccl3* and *Zfp36* mRNAs did not change. These data are consistent with the Datasets EV1 and EV4 which show binding to *Fos* and *Ccl12* mRNAs at 6 but not 3 h, whereas the other tested mRNAs are bound at both time points.

6) Minor point: "The authors state the highest number of TTP binding sites were located in introns (64%) and 3' UTRs (35%). The number of binding sites should be normalized to the length of intron and 3UTRs."

The intronic binding sites are narrow regions (i.e. not spread across the full length of intron) most often located around clustered AUUUA pentamers - to see this, we would like to invite the reviewer to search the TTP atlas (<http://ttp-atlas.univie.ac.at>) for genes exhibiting intronic TTP binding sites (Dataset EV4), e.g. *Il1a*, *Irg1*, *Cdk8* etc. We feel that normalization of PAR-iCLIP reads to the number of reads from control RNA-Seq experiment (i.e. using the same library preparation strategy) provides optimal weighting strategy. The normalization accounts for both the length of genomic features (introns, 3' UTR) and expression levels in those features. Using this approach, we observed that the normalized CLIP signal is nearly equally distributed between introns and 3' UTR. This result, which is now provided in the corresponding Results section (Results section 1, paragraph 5), further strengthens the finding that TTP frequently binds to introns.

7) Minor point: "The authors state that "this group represents mRNAs which might be regulated by competition of HuR and TTP for binding, similar as reported for *Tnf* mRNA (Tiedje et al, 2012)." In this context the reference Mukherjee et al. *Genome Biology* (2014) should be included."

The reference is now included (Results section 4, sentence 3).

Reviewer #2:

"The major problem with this paper is that PAR-CLIP is not a quantitative procedure. This is widely known. PAR-CLIP like other CLIP methods is very useful for mapping binding sites of most RNA binding proteins. However, the efficiency of UV crosslinking is poor and no publication has shown CLIP is able to saturate binding sites; in fact when has been measured saturation could not be demonstrated. This fact affects most of the experiments in this study; unfortunately, the field does not yet have a single technology that can solve the combinatorial problem quantitatively. Of course, this can be done using traditional biochemistry and structural analysis of purified RNAs and RNA binding proteins. Therefore, CLIP-seq techniques are not suitable for global analysis of biological dynamics like macrophage activation. Indeed, the Mukherjee 2014 paper was not able to directly solve the combinatorial problem quantitatively. Moreover, PAR-CLIP has significant background that would have to be accounted for prior to any rigorous quantitative normalization. Thus, without binding site coverage of the entire transcriptome and consideration of the authors cannot rigorously

measure the degree of overlap between TTP and HuR when using PAR-CLIP. Indeed, the publications to date regarding overlaps between TTP and HuR have a very wide variation in binding site assignments, as one would expect when binding sites are not saturated.”

We agree with the reviewer that “*PAR-CLIP like other CLIP methods is very useful for mapping binding sites of most RNA binding proteins*” and that “*the field does not yet have a single technology that can solve the combinatorial problem quantitatively*”. Along these lines, we would like to state, that one of the major aims of our study was to map TTP binding sites in immunostimulated macrophages and to functionally annotate them with regard to the stability and expression levels of the corresponding mRNAs. In agreement with the reviewer, the chosen approach was appropriate to achieve our goal.

Although the combinatorial effect of TTP and HuR is more difficult to address using the currently available technologies, for the following reasons we believe that our data represent valuable information:

- (i) We have observed binding of TTP and HuR among others to *Tnf* mRNA, i.e. mRNA which has been previously reported to be targeted by both proteins by using different techniques, including in vitro assays.
- (ii) We have now included an independent technique to show binding of TTP and HuR to the same targets – we have employed sequential native RNA-IP (**new Figure 4H**, described above in reply to the Reviewer 1, point 4). The sequential RNA-IP advances the previous knowledge since it demonstrates that TTP and HuR can bind simultaneously to target mRNAs.
- (iii) We have validated the CLIP data by native RNA-IPs (not dependent of cross-link efficiency) for 3 and 6 h of LPS treatment (**new Figure EV4**, described in reply to the Reviewer 1, point 5 and Reviewer 3, point 3). The native RNA-IPs show similar distribution of bound mRNAs at 3 and 6 h as determined by PAR-iCLIP.
- (iv) Quantitative normalization by transcript expression, calculated from RNA-Seq experiments, was conducted after peak finding and filtering to down-rank highly expressed transcripts with moderate crosslink signal compared to transcripts with high crosslink signal and moderate to low expression rates. The same procedure was followed for all replicates of TTP and HuR experiments.

2) *“Most conclusions in this manuscript are already claimed or demonstrated in other published studies. For example, many studies, including that of Hao and Baltimore have addressed changes in mRNAs encoding transcription factors and cytokines following macrophage activation. Many studies have shown that TTP controls TNF-alpha and other cytokine mRNAs but there is ambiguity in global reports regarding active versus silent targets of TTP, whether TTP mRNAs increase or decrease stability, etc. Overall, global TTP mRNA targeting experiments have shown mixed conclusions regarding what and how TTP targets subsets of mRNAs encoding cytokines and other immune regulatory factors, and this manuscript does not resolve the confusion in the field.”*

We feel that our study represents a major advance in the field because, among other reasons, it for the first time employs a physiological system: (i) macrophage is so far the only cell type in which TTP function has been established in vivo; (ii) primary macrophages express natural amounts of properly regulated TTP; (iii) immunostimulated macrophages express physiological TTP targets. In contrast, HEK293 cells used previously for TTP CLIP (Mukherjee et al. Genome Biology (2014)) lack natural TTP expression so that TTP overexpression was required. In addition, HEK293 cells barely express TTP targets. Although the HEK293 system is valuable for studying more general RNA binding proteins and for validation of techniques, this system is not suitable for studying the immune system-relevant TTP protein, as apparent from the fact that the TTP CLIP approach in HEK293 cells missed *Tnf* as well as the vast majority of other key TTP targets. Thus, for the understanding of the TTP function and for novel insights into the regulation of immune responses by TTP, binding site mapping in the macrophage transcriptome is indispensable.

We disagree with the reviewer’s statement “*there is ambiguity in global reports regarding active versus silent targets of TTP, whether TTP mRNAs increase or decrease stability*” since an analysis of active versus silent TTP sites has so far not been reported. Mukherjee et al. Genome Biology (2014) examined mRNA expression but not decay in HEK293 cells. In addition, to our knowledge there is no study showing that TTP can stabilize target mRNAs.

A major advance of our approach is the functional annotation of TTP binding sites. In particular, the annotation of TTP binding sites with respect to the stability of the corresponding mRNA, which cannot and has not been carried out in the HEK293 system, allows a direct assessment of the destabilizing function of TTP. We further believe that the searchable TTP atlas (<http://ttp->

atlas.univie.ac.at) which integrates binding site data with mRNA expression and decay, will provide a valuable and research-sparking resource for the scientific community.

Reviewer #3:

1) *“1) I must say the text is sometimes too technical and difficult to follow and I found myself reading a paragraph 3 or 4 times to fully understand what the authors wanted to say. In this regard, I missed a more didactic description of the results to guide scientists from other fields throughout the manuscript. Although I am familiar with PAR-iCLIP, a detailed description of the method will help to non-experts. I found especially hard the description of the results associated to figure 7 and 8, and I do believe that a more didactic approach would help to highlight the quality of this work.”*

We have attempted to make the text easier to follow. First, several parts containing various numbers (e.g. gene IDs, binding site positions, numbers of CLIP reads etc.) have been transferred to the Methods section. Second, parts of the Results sections describing PAR-iCLIP data, combinatorial binding of TTP and HuR (Figure 4), structural analysis of target mRNAs (Figure 5) and correlation analyses (Figures 7 and 8) have been rephrased. Third, too technical explanations have been transferred to the Methods part. We believe that the text can now be followed also by less specialized readers.

2) *“One of the most interesting aspects is the binding to fairly stable introns. This invokes sponge-like mechanisms to regulate TTP function rather than the opposite. I wonder whether these introns are indeed circular. I understand that unveiling a potential regulatory mechanism represents a whole project by itself, but some preliminary data in this regard would be of high interest.”*

We agree with the reviewer that the frequent binding of TTP to introns is an intriguing issue. We have attempted to knock-out or silence the debranching enzyme Dbr1 in order to increase the amount of intronic RNA in cells. We reasoned that this might lead to a more pronounced “sponge” function of the intronic TTP binding. Unfortunately, due to the low viability of Dbr1-deficient/hypermorph cells, we did not obtain conclusive data. A more sophisticated, i.e. regulated, system is needed to test the sponge hypothesis. Because of a lack of time we refrained from a further characterization of TTP binding to introns in the current study.

3) *“The difference between binding in the 3' UTR increases from 157 to 198. Is this difference superior to what one may expect from technical variance between replicates at each time point? If it is not an artefact, differential binding can be easily validated by crosslinking, IP and RT-qPCR of few mRNAs.”*

A similar comment was made by this reviewer on the criticism raised by the reviewer 3:

“What I would like to see is an experimental validation that the differential TTP binding to these transcripts is indeed occurring (and what is observed is not just experimental variation). This can be easily carried out by immunoprecipitating the protein in absence of RNase treatment and measuring the dynamically bound mRNAs by RT-qPCR. This would provide a strong argument (not dependent on crosslinking efficiency) about the dynamical behavior of TTP at the two different time points.”

As suggested by the reviewer, we have validated the PAR-iCLIP data for 3 and 6 h LPS by carrying out RNA-IPs followed by RT-qPCRs for several targets (**new Figure EV4**). The native RNA-IPs confirm the PAR-iCLIP data: TTP binding to *Fos* and *Ccl12* mRNAs strongly increased between 3 and 6 h of LPS treatment whereas binding to *Tnf*, *Ccl3* and *Zfp36* mRNAs did not change. These data are consistent with the Datasets EV1 and EV7 which show binding to *Fos* and *Ccl12* mRNAs at 6 but not 3 h, whereas the other tested mRNAs are bound at both time points.

Furthermore, as a novel approach, we used sequential RNA-IPs (**new Figure 4H**, described above in reply to the Reviewer 1, point 4) to prove combinatorial binding of TTP and HuR. The data not only confirm that TTP and HuR can bind to the same targets, but they also establish that TTP and HuR can bind simultaneously to mRNAs containing binding sites for both proteins.

4) *“Following (1), It's a bit confusing that TTP only affect stability of a limited number of transcripts in a TTP-dependent manner, but in figure 7-8, this proteins shapes the transcriptome. A more diligent explanation of the data would help to assess correctly all these data.”*

We have rephrased the text in the Results section 2 describing percentage of mRNAs destabilized by TTP, and replaced the wording “shapes the transcriptome” by “regulates” or “controls the transcriptome”. Furthermore, we have discussed in a more explicit way how TTP-dependent regulation of a few key inflammation drivers can determine the macrophage transcriptome.

5) Minor point “*What does it mean: "one replicate was used for further detailed analysis"? This doesn't sound right.*”

As explained in reply to the reviewer 1, point 1, the original manuscript text consistently describing our approach. The statements in the original Methods but not in the Results part were correct: a consensus binding site dataset has been generated from all replicates, and this dataset has been used throughout the study for all analyses. The incorrect parts in the Results text have been changed accordingly.

6) Minor point “*Figure 1A and 6A could be potentially biased due to differences in sequence length. i.e. if introns are in average longer than ORFs is not surprising to find enrichment of introns upon unspecific binding. The same applies for 5' and 3' UTRs. Normalisation of the data to sequence length can be used to estimate whether the binding in these regions is higher than the expected by random chance..*”

Our peak finding (Pyicos) and filtering approaches generate only high confidence binding sites and they efficiently reduce the background. The Pyicos modFDR method reliably distinguishes true CLIP signal from noise by comparing signal to background. For a detailed description of the algorithm and its performance, please refer to the original publication (Althammer et al., *Bioinformatics* 2011, 27: 3333-3340). In addition, we added stringent filters to obtain a list of peaks that not only show above background signal, but also have at least 100 crosslinks at their summit and are present in all three replicates. As a result, the identified binding sites are very narrow. To see this, we would like to refer to the TTP atlas (<http://ttp-atlas.univie.ac.at>) which visualizes the binding sites. As described in reply to Reviewer 1, point 6, we normalized PAR-iCLIP reads to the number of reads from control RNA-Seq experiment (i.e. using the same library preparation strategy). This approach provides an optimal weighting strategy since it accounts for both the length of genomic features (introns, 3' UTR) and expression levels in those features. Using this approach, we see that the normalized CLIP signal is nearly equally distributed between introns and 3' UTR. This result (now provided in the corresponding Results section 1, paragraph 5) further supports our binding site finding approach (Pyicos + filtering) and the data d in Figures 1A and 6A.

7) Minor point “*Authors mention that the PAR-CLIP determined sites correlate well in to the sites previously reported for Cxcl1, Illa and Cxcl2. I wonder if it could be possible to represent these data in a schematic. It could be quite illustrative about how these system-wide data fit into previous mechanistic studies.*”

A table showing previously reported positions of TTP binding sites in *Ccl3*, *Cxcl1*, *Cxcl2* and *Ill1a* mRNAs as well as the positions identified by PAR-iCLIP is now provided (**new Table 1**).

8) Minor point “*When correlating the binding profile to secondary structure and primary sequence I wonder why the authors used 7 nt (sliding?) windows. Stems can be potentially form from longer distances and I am afraid this analysis could miss a number of secondary structures. Perhaps I understood wrong this part, if so, some clarification would help.*”

We used RNAplfold to predict accessibility profiles for the whole mRNA sequence for each target of TTP/HuR. RNAplfold is a local folding algorithm and we employed a sliding window of length $w=75$ to compute the accessibility for stretches of 7nt. Thus, we computed the probability that a 7nt stretch would be unpaired within a 75nt window. The 7nt stretch was chosen, since the main AU-rich motif identified as TTP target in this dataset, WATTTAW, is 7nt long and we were interested in the probability of the whole motif being unstructured. We hope the revised text clarifies the distinction between sliding window size and the length of the accessible region. For the correlation analysis we extracted regions around known AU-rich elements that show binding (positive set) or no binding (negative set) of TTP/HuR, allowing us to compare accessibilities between bound and unbound ARE motifs.

2nd Editorial Decision

01 April 2016

Thank you again for submitting your work to Molecular Systems Biology. We have now heard back from the three referees who agreed to evaluate your manuscript. As you will see below, while reviewers #1 and #3 think that the revised manuscript is suitable for publication, reviewer #2 still raises a number of concerns.

We have circulated the reports to all reviewers as part of our 'pre-decision cross-commenting'

policy. During this process, reviewer #3 indicated that s/he thinks that the remaining issues can be addressed. S/he provides constructive comments, which we paste below for your reference (see "Reviewer #2: additional remarks after cross-commenting"), since we think they can be rather helpful for revising the manuscript.

REFEREE REPORTS

Reviewer #1:

The author have adequately addressed the reviewer's concerns.
The manuscript should be considered for publication in MSB.

Reviewer #2:

I am afraid that the authors were less than responsive to my comments and questions such that my confidence in this paper is diminished. The authors responded mostly by reiterating the conclusions in the text as "valuable information". I find much of that information to be incremental and/or already presented in other publications regarding TTP and HuR. For example, many reports, including the original animal genetics of Blackshear et al. have shown that TNF α mRNA is a target of TTP. The statement that some papers did not find TNF α mRNA to be a target of TTP is not a meaningful conclusion, other than to lead the reader to doubt the methods used. Likewise, while the sequential native IP appears to work well, the finding that TTP and HuR can bind to the same mRNA using sequential native IP (Figure 4H) is not novel and does not significantly "advance previous knowledge". In addition, I share the confusion noted by other reviewers regarding the analytical approach and interpretation. In fact, upon diving deeper into the methods section, I find it difficult to understand how the experimental design and evidence supports many of the authors' conclusions.

1. Most important to this reviewer is that the analysis of the PAR-iCLIP data is atypical and very difficult to understand. The finding of so few genes with TTP or HuR binding sites is not typical of other published PAR-CLIP or CLIP procedures. Others tend to use algorithms that set the parameters (e.g. PARalyzer and others). Perhaps the limited number of genes found is explained by the limited depth of sequencing being (TTP ~17M; HuR ~38M). Also, excluding TTP targets that have both intronic and 3'UTR binding is inexplicable and appears to be arbitrary. Moreover, the cutoff decisions are confusing. For example, they seem to be based on examining "known TTP targets" (page 20) and that introduces a major bias. Also, the use of 100 pileups for a cutoff is exceedingly high considering that PAR-CLIP generally uses a cutoff of 5.
2. In figure 2, the data show that 71% of the 3'UTR targets are stable. That is a striking number and likely to not be correct. Also, it is important, if not essential, to measure the changes in stability of non-targets as well as targets.
3. The data of figure 6C and 7B are based on numbers (%) of genes (rather than p values) in each GO category. And I do not understand how there can be zero values at 3 and 6 hours for broad GO categories such as "regulation of biological process" or "regulation of cellular process". This makes no sense.
4. I find the normalization data to be uninterpretable. Comparing the Y axes scales used in figures 8 A and B, the left panels would shrink into the right panels and compress the scatter of both left panels. There are no lines extrapolatable because the data are scattered in the LPS3 h and flat in the LPS6h panels. I do not see how those data are meaningful as metrics for normalization. Again, my original criticism was that CLIP procedures are inefficient and not quantitative; and I still believe the authors' results demonstrate precisely that. The advantage claimed by the authors is that this study "...for the first time employs a physiological system...". But comparison of "physiological" animal cells with primary macrophase and other cell lines is only useful if entirely novel results and well supported data are derived. I believe that the conclusions in this study are not well supported by the data presented.

Reviewer #3:

Sedlyarov et al., have answered all my concerns in the revised version of this manuscript. The

strength of this work is the combination of a physiological system (BMDM) and the state-of-the-art RNA biology methods to explore the biological role of TTP. In the revised version, the authors provide additional validation experiments to confirm the most relevant results such as native RNA-IP of differentially bound transcripts at 3 or 6 h post LPS treatment. I have also noticed that the manuscript reads better and it is easier to follow. Overall, I am satisfied with the point by point clarifications and the additional experiments provided.

Reviewer #2: additional remarks after cross-commenting:

Regarding the small number of TTP-bound genes, the large percentage (71%) of stable TTP targets and the data shown in Figure 8:

-> From my point of view we find in science dogmas about the "well described examples". Basically, a function derived from an individual example (e.g. TTP regulating stability of TNF), often becomes global (i.e. TTP regulates RNA stability). High throughput data is now giving us a broader perspective regarding functional protein-RNA interactions and I am not surprised that TTP binding may result in different outcomes, probably depending on the configuration of RBPs within a given RNP. HuR represents just one example of potential cross-regulation of TTP, and there are about 1300 proteins currently annotated as RBPs. The outcome of such diverse interactions could be unpredictable. What is clear from this work is that TTP interaction with RNA appears not to be "sufficient" to induce RNA degradation. Because this result adds a new layer on information on TTP function, I would suggest to highlight this as a major and novel conclusion in the manuscript.

For example, many reports, including the original animal genetics of Blackshear et al. have shown that TNF mRNA is a target of TTP.

-> I don't think the authors claim anything about this result, apart from the validation of their PAR-CLIP data.

Likewise, while the sequential native IP appears to work well, the finding that TTP and HuR can bind to the same mRNA using sequential native IP (Figure 4H) is not novel and does not significantly "advance previous knowledge". In addition, I share the confusion noted by other reviewers regarding the analytical approach and interpretation. 

-> I interpret this figure as validation of the PAR-CLIP data, which is valuable. I would thus suggest the authors to highlight this aspect in the text.

1. Most important to this reviewer is that the analysis of the PAR-iCLIP data is atypical and very difficult to understand. The finding of so few genes with TTP or HuR binding sites is not typical of other published PAR-CLIP or CLIP procedures. Others tend to use algorithms that set the parameters (e.g. PARalyzer and others). Perhaps the limited number of genes found is explained by the limited depth of sequencing being (TTP ~17M; HuR ~38M).

-> These numbers of reads are normal in PAR-CLIP and iCLIP data. Higher number of reads would make me suspicious of major contaminations.

Also, excluding TTP targets that have both intronic and 3'UTR binding is inexplicable and appears to be arbitrary.

-> I agree with this, but this is easy to modify.

Moreover, the cutoff decisions are confusing. For example, they seem to be based on examining "known TTP targets" (page 20) and that introduces a major bias. Also, the use of 100 pileups for a cutoff is exceedingly high considering that PAR-CLIP generally uses a cutoff of 5.

-> The authors could provide supplementary information with sites with lower read coverage as "candidate sites" in the tables/website. Nevertheless I understand why the authors increased the threshold to 100 reads, since, as far as I know, PAR-CLIP primers lack random barcodes and cannot

exclude over-amplification rounds in the PCR reaction. Therefore, 5 reads doesn't sound right to me since they can be derived from one single RNA molecule.

2. In figure 2, the data show that 71% of the 3'UTR targets are stable. That is a striking number and likely to not be correct. Also, it is important, if not essential, to measure the changes in stability of non-targets as well as targets.

-> I don't think unbiased/comprehensive approaches should sound right or wrong if experimentally solid. The PAR-CLIP dataset is well done and my conclusion is that TTP binding can induce different outcomes, probably depending of the protein partners associated with it. I would indeed highlight this conclusion if the numbers are right (i.e. 71%).

3. The data of figure 6C and 7B are based on numbers (%) of genes (rather than p values) in each GO category. And I do not understand how there can be zero values at 3 and 6 hours for broad GO categories such as "regulation of biological process" or "regulation of cellular process". This makes no sense.

-> This might be wrong and it is easy to revise.

4. I find the normalization data to be uninterpretable. Comparing the Y axes scales used in figures 8 A and B, the left panels would shrink into the right panels and compress the scatter of both left panels. There are no lines extrapolatable because the data are scattered in the LPS3 h and flat in the LPS6h panels. I do not see how those data are meaningful as metrics for normalization. Again, my original criticism was that CLIP procedures are inefficient and not quantitative; and I still believe the authors' results demonstrate precisely that. The advantage claimed by the authors is that this study "...for the first time employs a physiological system...". But comparison of "physiological" animal cells with primary macrophase and other cell lines is only useful if entirely novel results and well supported data are derived. I believe that the conclusions in this study are not well supported by the data presented.

-> I am not surprised that data from very different kind of experiments show a moderate correlation, but it isn't bad here. Maybe the authors can think about an alternative manner to display these data in a more intuitive manner. I may suggest to tone down this part, authors could move these panels to the supplemental figures or fuse it to figure 7. Alternatively, they can tone down the text referring to these plots.

2nd Revision - authors' response

07 April 2016

Point-by-point reply to the comments of reviewer 2:

(reviewer #2 comments in italics; reviewer #3 suggestions/comments in bold italics)

1)

Reviewer 2: *"I find much of that information to be incremental and/or already presented in other publications regarding TTP and HuR. For example, many reports, including the original animal genetics of Blackshear et al. have shown that TNF mRNA is a target of TTP."*

Reviewer 3:

- ***" -> I don't think the authors claim anything about this result, apart from the validation of their PAR-CLIP data. "***
- ***"What is clear from this work is that TTP interaction with RNA appears not to be "sufficient" to induce RNA degradation. Because this result adds a new layer on information on TTP function, I would suggest to highlight this as a major and novel conclusion in the manuscript. "***
- ***"The strength of this work is the combination of a physiological system (BMDM) and the state-of-the art RNA biology methods to explore the biological role of TTP."***

Re: We would like to emphasize that our work is the first study reporting (i) genome-wide mapping of TTP binding sites in a physiological system (primary macrophages), (ii) functional annotation of

physiological TTP binding sites, (iii) comparison of TTP and HuR binding sites in macrophages. Only such a comprehensive analysis allows insights into TTP function. One of the conclusions that can be drawn is that TTP binding is not sufficient to induce physiological target degradation. This view is explicitly supported by the reviewer 3. Given the positive evaluation by reviewer 1, we believe that reviewer 1 shares this opinion as well.

As for the statement on Tnf, we use this TTP target solely for the validation of our approach. This is explicitly stated at several occasions in the manuscript. E.g. in the first part of the Results section we say: “This is exemplified by TTP binding to the Tnf transcript: TTP binding site was centered at the nucleotide (nt) 2,315 in the transcript (corresponding to nt 1,326 in mRNA) in all replicates (Figure 1 and EV2B), in agreement with the reported TTP binding to Tnf reporter constructs (Lai et al, 1999)”. We appropriately reference the seminal study published by the Blackshear lab on TTP binding to Tnf mRNA so that it is clear that the Tnf data in our work is not novel. The supportive statement of the reviewer 3 indicates that we have correctly described the Tnf data in the manuscript.

We have followed the suggestion of the reviewer 3 to highlight the finding the TTP binding is not sufficient for target destabilization by including an appropriate statement in the abstract.

2)

Reviewer 2 “*The statement that some papers did not find TNF α mRNA to be a target of TTP is not a meaningful conclusion, other than to lead the reader to doubt the methods used.*”

Re: We assume that the statement of the reviewer is referring to: “Notably, Tnf was not found among the targets in HEK293 cells, even though it is the most common TTP target in various cells. Other cytokine mRNAs were similarly absent (Table EV1). These dissimilarities might be caused by TTP overexpression, and hence a reduced binding specificity, in the HEK293 system. Furthermore, HEK293 cells are not immune cells and barely express genes related to the immune response. They also lack TTP activity regulated by immune cell signaling. Thus, the paucity of natural TTP targets in HEK293 cells and/or different TTP regulation might cause binding of TTP to RNAs different from those targeted in immune cells.” We don’t feel that such statement doubts other methods or that it is a conclusion. Rather, the sentence describes several important facts and provides an explanation for differences between our results and the reported findings. Thus, we regard it as an explanatory text which should help the readers to better understand the study.

3)

Reviewer 2: “*Likewise, while the sequential native IP appears to work well, the finding that TTP and HuR can bind to the same mRNA using sequential native IP (Figure 4H) is not novel and does not significantly "advance previous knowledge".*”

Reviewer 3: “**-> I interpret this figure as validation of the PAR-CLIP data, which is valuable. I would thus suggest the authors to highlight this aspect in the text.**”

Re: To the best of our knowledge, sequential IPs for endogenous TTP and HuR have not been reported. If the reviewer 2 has different information, we would appreciate if the study showing sequential IPs was referenced by the reviewer. The reviewer 3 supports our view and suggests to highlight this data as an important validation experiment. We have modified the text accordingly.

4)

Reviewer 2: “*1. Most important to this reviewer is that the analysis of the PAR-iCLIP data is atypical and very difficult to understand. The finding of so few genes with TTP or HuR binding sites is not typical of other published PAR-CLIP or CLIP procedures. Others tend to use algorithms that set the parameters (e.g. PARalyzer and others). Perhaps the limited number of genes found is explained by the limited depth of sequencing being (TTP ~17M; HuR ~38M).*”

Reviewer 3: “**-> These numbers of reads are normal in PAR-CLIP and iCLIP data. Higher number of reads would make me suspicious of major contaminations.**”

Re: We employed a slightly modified protocol of CLIP described by the Jernej Ule lab in Nat Struct Mol Biol. 2010 Jul;17(7):909-15 and Methods. 2014 Feb;65(3):274-87. The modifications arose from the necessity to adjust the method to our cell system and antibodies. For the analysis of binding sites we used Pyicos (Bioinformatics. 2011 Dec 15;27(24):3333-40.) which has been developed for

analysis of high-throughput sequencing data, including CLIP, as explicitly stated in the abstract of the Pyicos paper.

As for the number of reads, which is considered by the reviewer to be low, we would like to point to a very recent review of published CLIP experiments (Nucleic Acids Res. 2015 Jun 23;43(11):5263-74). The table 2 in that paper shows that the number of mapped reads in our experiments is well above average. In fact, some of the published CLIP experiments contained less than 1/10 of the read number obtained in our study.

Together, our approach, which represents one of several possible ways of doing CLIP experiments, is supported by reported studies and can be regarded as correct. This view is shared by the reviewer 3.

5)

Reviewer 2: *“Also, excluding TTP targets that have both intronic and 3'UTR binding is inexplicable and appears to be arbitrary.”*

Reviewer 3: *“ -> I agree with this, but this is easy to modify.”*

Re: We believe that the reviewer is referring to the Figure 2A and 2B showing stability of 3' UTR and intronic TTP targets. In this analysis we did not exclude genes with both 3' UTR and intronic binding: genes exhibiting binding in both regions are included in both Figure 2A and 2B. This is also the reason why the sum (541 genes) of genes in 2A (198 genes) and 2B (343 genes) is higher than the total number of TTP target genes identified and shown in the Dataset EV1 (498). We have clarified this in the main text and figure legend.

6)

Reviewer 2: *“Moreover, the cutoff decisions are confusing. For example, they seem to be based on examining “known TTP targets” (page 20) and that introduces a major bias. Also, the use of 100 pileups for a cutoff is exceedingly high considering that PAR-CLIP generally uses a cutoff of 5.”*

Reviewer 3: *“-> The authors could provide supplementary information with sites with lower read coverage as “candidate sites” in the tables/website. Nevertheless I understand why the authors increased the threshold to 100 reads, since, as far as I know, PAR-CLIP primers lack random barcodes and cannot exclude over-amplification rounds in the PCR reaction. Therefore, 5 reads doesn't sound right to me since they can be derived from one single RNA molecule.”*

We agree that cutoffs are arbitrary decisions. In our study we set the cutoffs such that only high confident sites were defined as TTP binding sites. The good agreement of our dataset with the so far reported information about TTP binding sites (mentioned in the text and shown in Figure EV2 for Tnf, and in Table 1 for other targets) indicates that our strategy was in principle correct. Our TTP atlas website comprises the full dataset and visualizes all detected TTP PAR-CLIP reads at any annotated gene expressed in macrophages. Thus, the readers can quickly obtain an idea about TTP binding to the gene of interest, and they can download the dataset and modify the filtering rules to their needs. We intend to implement additional features in this website once we start obtaining feedback from the users.

7)

Reviewer 2: *“2. In figure 2, the data show that 71% of the 3'UTR targets are stable. That is a striking number and likely to not be correct. Also, it is important, if not essential, to measure the changes in stability of non-targets as well as targets.”*

Reviewer 3: *“ -> I don't think unbiased/comprehensive approaches should sound right or wrong if experimentally solid. The PAR-CLIP dataset is well done and my conclusion is that TTP binding can induce different outcomes, probably depending of the protein partners associated with it. I would indeed highlight this conclusion if the numbers are right (i.e. 71%).”*

Re: We cannot do else than refer to the supportive statement of the reviewer 3. Data obtained using unbiased high content approaches should not be judged as correct or wrong if the experiments are sound, include appropriate controls and are validated using independent approaches. Whether one likes such results or not, these data are in agreement with increasing evidence for more complex roles of RNA binding proteins as revealed for example by comprehensive genome-wide analyses of

HuR (Mol Cell. 2011 Aug 5;43(3):327-39.) (Cell Rep. 2014 Dec 24;9(6):2330-43.) (Nat Immunol. 2015 Apr;16(4):415-25.).

As for the request to measure the stability of targets as well as non-targets, we would like to refer to the Dataset EV2 – this dataset as well as the manuscript text show that we indeed have analyzed the stability of the entire macrophage transcriptome. The Dataset EV2 contains the complete dataset for 3 and 6 h of LPS treatment of WT and TTP-deficient macrophages.

8)

Reviewer 2: "3. The data of figure 6C and 7B are based on numbers (%) of genes (rather than p values) in each GO category. And I do not understand how there can be zero values at 3 and 6 hours for broad GO categories such as "regulation of biological process" or "regulation of cellular process". This makes no sense."

Reviewer 3: " -> **This might be wrong and it is easy to revise.**"

Re: These figures show only GO categories found to be significantly enriched in a GO-enrichment analysis. Specifically, we present only significantly (adj. p-value < 0.05) enriched GO categories. This fact might not have been clearly explained in the manuscript. We have now included two more tables showing statistics of the enrichment analysis: Table EV3 (related to Figure 6C; TTP binding at 3 and 6 h of LPS treatment) and Table EV4 (related to Figure 7B; differential expression at 3 and 6 h of LPS treatment). Furthermore, we have included a more explicit description both in the main text, figure and figure legend.

9)

Reviewer 2: "4. I find the normalization data to be uninterpretable. Comparing the Y axes scales used in figures 8 A and B, the left panels would shrink into the right panels and compress the scatter of both left panels. There are no lines extrapolatable because the data are scattered in the LPS3 h and flat in the LPS6h panels. I do not see how those data are meaningful as metrics for normalization."

Reviewer 3: " -> **I am not surprised that data from very different kind of experiments show a moderate correlation, but it isn't bad here. Maybe the authors can think about an alternative manner to display these data in a more intuitive manner. I may suggest to tone down this part, authors could move these panels to the supplemental figures or fuse it to figure 7. Alternatively, they can tone down the text referring to these plots.** "

Re: We have rescaled the plots on the Figure 8 so that all 4 panels are now in the same scale, as requested by the reviewer. We believe that this modification helps visualizing the correlation particularly in Figure 8A (differential decay versus TTP binding score). Nevertheless, the key parameter in all panels in this figure is the Pearson correlation coefficient. The coefficient shows that correlation increases from 3 to 6 h of LPS treatment. Importantly, there is no correlation between differential expression versus TTP binding at 3 h of LPS treatment (Figure 8B, left panel: p-value = 0.36), whereas there is a significant correlation in all other panels (p-value < 2×10^{-16}). An obvious complication in visualizing the correlation comes from the high number of TTP-bound but stable mRNAs which create noise on the plots. To make this analysis easier to understand, we have emphasized the correlations and their significance in the main text, replaced the "r" symbol for the Pearson coefficient by the more frequently used "r" in the panels, and included confidence intervals in the figure legend.

As for the statement "I do not see how those data are meaningful as metrics for normalization", we assume that the reviewer is referring to the title of the panels ("normalized binding score"). We do not use it as metrics for normalization. Rather, the normalized binding score defines TTP binding site scores normalized to the expression of target mRNAs (obtained from RNA-Seq data). Such normalization allows relative comparison of TTP binding strength to different targets. This is described in the main text pertaining to the Figure 8 as well as in the Methods section.

10)

Reviewer 2: "Again, my original criticism was that CLIP procedures are inefficient and not quantitative; and I still believe the authors' results demonstrate precisely that. The advantage claimed by the authors is that this study "...for the first time employs a physiological system...". But comparison of "physiological" animal cells with primary macrophage and other cell lines is only

useful if entirely novel results and well supported data are derived. I believe that the conclusions in this study are not well supported by the data presented. “

Re: PAR-CLIP and related methods (e.g. HITS-CLIP, PAR-iCLIP), pioneered by the Darnell, Tuschl, Ule and Zavolan labs, are considered as state-of-the-art techniques for quantitative as well as qualitative analysis of RNA-protein interactions (Nature. 2008 Nov 27;456(7221):464-9.) (Cell. 2010 Apr 2;141(1):129-41.) (Nat Struct Mol Biol. 2010 Jul;17(7):909-15.) (Nat Methods. 2011 May 15;8(7):559-64.). There will certainly be future developments which will further improve the analysis of RNA-proteins interactions, but the PAR-CLIP techniques are currently the most appropriate, as evident from a number of influential studies. The technique is still quite challenging if attempting to analyze endogenous (e.g. not overexpressed and tagged) proteins in an environment which is close to in vivo situation. For this reason, our study which employs a combination of a physiological system with PAR-CLIP analysis represents a major advance in the field, as indicated by the reviewers 1 and 3.

Corresponding Author Name: Pavel Kovarik

Manuscript Number: MSB-15-6628